# Evaluation of molecular subtypes and clonal selection during establishment of patient-derived tumor xenografts from gastric adenocarcinoma

Anne-Lise Peille[1,9,12], Vincent Vuaroqueaux [1,9,12], Swee-Seong Wong [2,10], Jason Ting[2], Kerstin Klinger[1], Bruno Zeitouni[1], Manuel Landesfeind[1,11], Woo Ho Kim [3], Hyuk-Joon Lee[4], Seong-Ho Kong[5], Isabella Wulur[2], Steven Bray[2,10], Peter Bronsert [6,7,8], Nina Zanella[1], Greg Donoho[2], Han-Kwang Yang[4], Heinz-Herbert Fiebig [1,9,13✉], Christoph Reinhard[2,13✉] & Amit Aggarwal [2,13✉]

Patient-derived xenografts (PDX) have emerged as an important translational research tool for understanding tumor biology and enabling drug efficacy testing. They are established by transfer of patient tumor into immune compromised mice with the intent of using them as Avatars; operating under the assumption that they closely resemble patient tumors. In this study, we established 27 PDX from 100 resected gastric cancers and studied their fidelity in histological and molecular subtypes. We show that the established PDX preserved histology and molecular subtypes of parental tumors. However, in depth investigation of the entire cohort revealed that not all histological and molecular subtypes are established. Also, for the established PDX models, genetic changes are selected at early passages and rare subclones can emerge in PDX. This study highlights the importance of considering the molecular and evolutionary characteristics of PDX for a proper use of such models, particularly for Avatar trials.

[1] Charles River Discovery Research Services Germany GmbH (formerly Oncotest GmbH), Am Flughafen 12-14, 79108 Freiburg, Germany. [2] Lilly Research Labs, Eli Lilly and Company, Indianapolis, IN 46285, USA. [3] Department of Pathology, Seoul National University College of Medicine, 101 Daehak-ro, Jongno-gu, Seoul, Korea. [4] Department of Surgery and Cancer Research Institute, Seoul National University College of Medicine, 101 Daehak-ro, Jongno-gu, Seoul, Korea. [5] Department of Surgery, Seoul National University Hospital, 101 Daehak-ro, Jongno-gu, Seoul, Korea. [6] Institute for Surgical Pathology, Medical Center—University of Freiburg, Freiburg, Germany. [7] Comprehensive Cancer Center Freiburg, Medical Center—University of Freiburg, Freiburg, Germany. [8] Faculty of Medicine, University of Freiburg, Freiburg, Germany. [9]Present address: 4HF Biotec GmbH, Am Flughafen 14, Freiburg 79108, Germany. [10]Present address: LifeOmic, 351 W 10th St, Indianapolis, IN, USA. [11]Present address: Evotec International GmbH, Marie-Curie-Strasse, 37079 Göttingen, Germany. [12]These authors contributed equally: Anne-Lise Peille, Vincent Vuaroqueaux. [13]These authors jointly supervised this work: Heinz-Herbert Fiebig, Christoph Reinhard, Amit Aggarwal. ✉email: fiebig@4hf.eu; reinhard_christoph@lilly.com; aggarwal_amit@lilly.com

In the context of drug development, there is a continued need of preclinical models well covering the key aspects of the disease biology. With ability to propagate human tumor materials in immune-compromised mice, patient-derived xenografts (PDX) have increasingly become a cornerstone of anticancer agent testing. Whereas these models were shown to well mimic response to therapeutics[1–7], recent studies pointed out the needs for large PDX collections to capture the cancer heterogeneity[8]. Previous studies focusing on gastric cancer PDX models, reported the preservation of the parental tumor histology in these models as well as their stability over passages. However, a low engraftment success rate with histology subtype selection was also often observed suggesting bias in these formed collections[7,9–12].

Extensive molecular characterization of gastric cancer has revealed cancer heterogeneity due to diverse etiological factors and genetic mechanisms underlying its pathogenesis. The cancer genome atlas (TCGA[13]) project recently highlighted the landscape of genomic alterations in gastric cancer and proposed to classify tumors into four molecular subtypes: tumors with microsatellite instability (MSI), Epstein-Barr virus (EBV), chromosome instable tumors (CIN), and tumors with genomic stability (GS). Asian cancer research group (ACRG[14]) established another classification based on tumor transcriptomic profile. Gastric cancers were divided into MSI and microsatellite stable (MSS) with MSS tumors further divided into MSS/EMT and MSS/TP53+ and MSS/TP53− subtypes representing epithelial–mesenchymal transited, activation or inactivation of the TP53 pathway, respectively.

The goals of this study were to establish a collection of Asian gastric cancer PDX using a patient gastric cancer cohort shown to be representative for key clinicopathologic features and to determine if the molecular subtypes and heterogeneity of the established PDX models is adequately represented in patient tumors. For this, biologic materials were collected at the different steps of the PDX establishment process to conduct extensive genomic and transcriptomic analyses. We investigated whether heterogeneity as embodied by clonality, genetic alterations, and molecular subtypes is retained and if any biases are introduced by gastric cancer PDX establishment process.

## Results

### Representativeness of the patient tumors used for PDX establishment.
We received resected tumor materials from $n = 100$ primary gastric cancer patients of Asian ethnicity from Seoul Hospital for PDX establishment (2008–2014). Study design is described in the "Methods" section and detailed clinical data are shown in Table 1 and Supplementary Data 1. We first checked whether our cohort well-covered key histopathological and molecular subtypes of gastric cancer tumors. We observed that the distribution for patient gender, tumor location, WHO grades, Lauren[15] subtypes, lymph node invasion, and metastasis were comparable to those recently reported by the ACRG and TCGA studies[13,14]. At molecular level, our cohort contains a similar proportion of key ACRG[14] and TCGA[13] molecular subtypes (Fig. 1 and Supplementary Data 2). Pentaplex microsatellite assay identified 27/100 MSI-positive tumors (common subtype to both ACRG and TCGA) with high MSI. According to the ACRG, 13/100 tumors were classified as MSS/EMT, 32/100 as MSS/TP53− and 21/100 as MSS/TP53+ using qRT-PCR (Supplementary Data 3). Seven MSS tumors were not classified due to a lack of RNA. Regarding TCGA classification, 10 tumors were EBV positive with high Epstein Barr virus (EBV) infection burden (quantification done by qPCR) and we could not ascertain the GS or CIN TCGA subtypes for the remaining 63 tumors as it needed an Affymetrix SNP6.0 assay or equivalent.

In agreement with previous work[13,14], the ACRG classification of the 100 tumors was associated with tumor location (Chi-square test, $p = 0.001$), tumor stage (Chi-square test, $p = 0.0013$), lymph node stage (Chi-square test, $p = 0.005$), distant metastasis stage (Chi-square test, $p = 0.009$), and the lymphovascular (Chi-square test, $p = 0.001$), venous (Chi-square test, $p = 0.002$) and perineural (Chi-square test, $p < 0.0001$) invasion (Supplementary Data 4a). MSI tumors were mainly of the intestinal Lauren subtype (ACRG, Chi-square test, $p = 0.0025$; TCGA (MSI vs. EBV), Chi-square test, $p = 0.066$), frequently observed in the lower third part of the stomach (ACRG: Chi-square test, $p = 0.001$, TCGA: Chi-square test, $p = 0.021$) and presented a less frequent perineural invasion in contrast to the other groups (ACRG, Chi-square test, $p = 0.0001$; TCGA Chi-square test, $p = 0.036$). MSS/TP53+ tumors were frequently EBV-infected (Chi-square test, $p < 0.0001$) and of the diffuse subtype (as EMT tumors), whereas MSS/TP53− tumors were dominantly intestinal (Chi-square test, $p = 0.023$). By considering TCGA classification, EBV tumors were dominantly diffuse (MSI vs. EBV, Chi-square test, $p = 0.066$) and preferentially localized in the upper third part of the stomach (Chi-square test, $p = 0.021$). The MSS (CIN or GS) tumors were more frequently retrieved in male patients (Chi-square test, $p = 0.014$) (Supplementary Data 4b).

### PDX were not established from all molecular subtypes.
From the 100 gastric tumors implanted in NMRI nude mice, a total of $n = 27$ PDX models stably growing over passage 4 were validated as PDX models (see "Methods" section). The period between initial tumor implantation and first passage (P1) ranged from 1 week to 41 weeks (mean 15 weeks) without significant association with any histopathological parameter, but significantly associated with the mutation prevalence (Spearman correlation $r = −0.42$, $p = 0.035$) and the ACRG and TCGA molecular subtypes with MSI PDX growing faster than the MSS PDX (Kruskal–Wallis test, $p = 0.009$, Supplementary Fig. 1). As previously reported[14,16], PDX were more frequently established from intestinal than diffuse or mixed tumors (Chi-square test, $p = 0.008$, Fig. 1, Table 1). We showed in addition that PDX establishment is dependent of tumor molecular subtypes. In line with previous studies[17,18], the PDX collection was enriched in models established from MSI tumors ($n = 15/27$, Fisher's exact test, $p = 0.0002$, Table 1). A lower number of PDX were developed from MSS/TP53− ($n = 9/32$) and MSS/TP53+ (3/21), but none from EMT and EBV-positive tumors. Further analysis indicated that Lauren subtypes as well as ACRG and TCGA subtypes of PDX and respective parental tumors well matches together (Lauren: Chi-square test, $p = 0.0001$, 85% of match, ACRG: Chi-square test $p = 0.0003$ and TCGA: Fisher's exact test MSI vs. MSS $p < 0.0001$, Fig. 1, Supplementary Data 5 and 6). In more detail, we confirmed that PDX retained typical genomic alteration characteristics defining their subtypes. MSI tumors ($n = 15$) showed typical mismatch repair deficiency characteristics, such as alterations in MLH1 gene through mutation or loss of expression (Fig. 2), an hypermutation (MSI: 21.9–57.5 mutations/Megabase, mean = 32.9; MSS: 2.2–21.1 mutations/Megabase, mean = 7.95, Mann–Whitney test, $p < 0.0001$), a high proportion of indels (30% in MSI vs. 19% in MSS, Mann–Whitney test, $p = 0.0037$) and specific trinucleotide substitutions (Supplementary Fig. 2a, Supplementary Data 7). By applying the signatures of mutational processes defined by Alexandrov et al.[19], we observed that MSI PDX showed a high prevalence of the signatures 6 (mismatch repair deficiency), 1A and 1B (patient age), 12 and the gastric cancer-specific signatures 15 and 21 (Supplementary Fig. 2b). The MSS model GXA_3084, presenting a R494Q mutation in POLE, had a typical MSS mutation processes signature but was also hypermutated.

**Table 1 Patient and PDX characteristics and engraftment rates.**

| Variables | Asian gastric tumors | Asian gastric PDX | | Significance |
|---|---|---|---|---|
| | Number (n = 100) & frequency (%) | Number (n = 27) & frequency (%) | Success rate (%) | |
| Median age | 64 (37–90) | 65 (39–85) | | Mann–Whitney $p = 0.57$ |
| Gender | | | | Fisher´s exact test $p = 1$ |
| Female | 36 (36%) | 10 (37%) | 28 | |
| Male | 64 (64%) | 17 (63%) | 27 | |
| Lauren classification | | | | Chi-square test $p = 0.008$ |
| Intestinal | 53 (53%) | 21 (78%) | 40 | |
| Diffuse | 42 (42%) | 6 (22%) | 14 | |
| Mixed | 5 (5%) | 0 (0%) | 0 | |
| pT - primary tumor stage | | | | Chi-square test $p = 0.12$ |
| T1 | 6 (6%) | 0 (0%) | 0 | |
| T2 | 49 (49%) | 18 (67%) | 37 | |
| T3 | 32 (32%) | 8 (30%) | 25 | |
| T4 | 10 (10%) | 1 (4%) | 10 | |
| Unknown | 3 (3%) | 0 (0%) | 0 | |
| pN - lymph node stage | | | | Chi-square test $p = 0.68$ |
| N0 | 25 (25%) | 8 (30%) | 32 | |
| N1 | 30 (30%) | 9 (33%) | 30 | |
| N2 | 18 (18%) | 3 (11%) | 17 | |
| N3 | 26 (26%) | 7 (26%) | 27 | |
| Unknown | 1 (1%) | 0 (0%) | 0 | |
| pM - distant metastasis stage | | | | Chi-square test $p = 0.58$ |
| pM0 | 89 (89%) | 25 (93%) | 28 | |
| pM1 | 10 (10%) | 2 (7%) | 20 | |
| Unknown | 1 (1%) | 0 (0%) | 0 | |
| MSI status | | | | Fisher's exact test $p = 0.0002$ |
| MSI | 27 (27%) | 15 (56%) | 56 | |
| MSS | 73 (73%) | 12 (44%) | 16 | |
| Unknown | 0 (0%) | 0 (0%) | 0 | |
| ERBB2 IHC score | | | | Chi-square test $p = 0.57$ |
| 0 | 37 (37%) | 9 (33%) | 24 | |
| 1 | 34 (34%) | 8 (30%) | 24 | |
| 2 | 8 (8%) | 2 (7%) | 25 | |
| 3 | 7 (7%) | 5 (19%) | 71 | |
| Unknown | 14 (14%) | 3 (11%) | 21 | |
| EBV | | | | Chi-square test $p = 0.1$ |
| Positive | 10 (10%) | 0 (0%) | 0 | |
| Negative | 89 (89%) | 27 (100%) | 30 | |
| Unknown | 1 (1%) | 0 (0%) | 0 | |

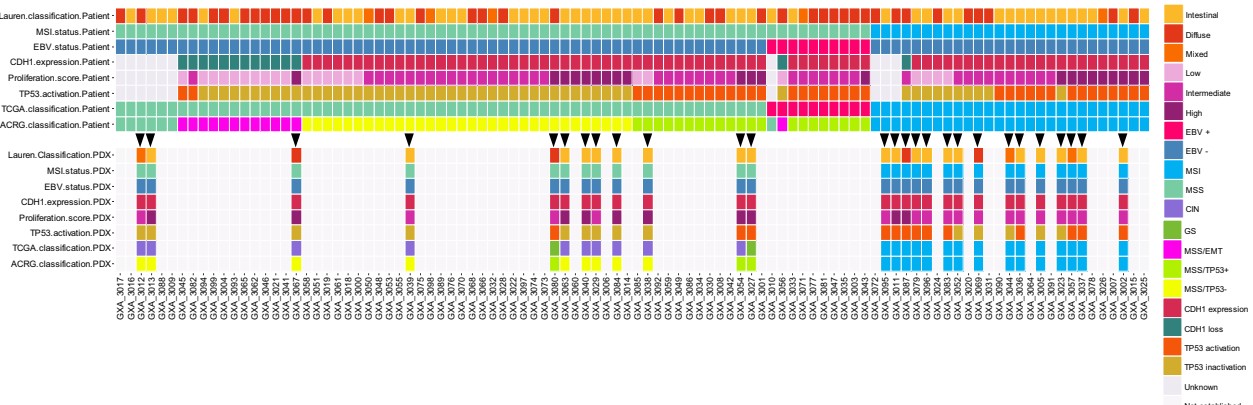

**Fig. 1 Subtypes of gastric cancer tumors and resulting PDX.** Patient tumors and PDX were investigated for histology and both ACRG (MSI, MSS/EMT, MSS/TP53+ and MSS/TP53−) and TCGA (MSI and EBV, the remaining MSS tumors were considered as GS or CIN) molecular subtypes. MSI status was determined using the pentaplex PCR (patient tumors) or by MLH1 loss as surrogate marker for the PDX (see Supplementary Fig. 1). MSS/EMT signature was determined by assessing the *CDH1* mRNA expression, and the *TOP2A*, *MKI67* mRNA expression as markers of proliferation by quantitative real-time PCR. The remaining MSS tumors were classified as MSS/TP53+ and MSS/TP53− using a two-gene *TP53* activation signature (*MDM2* and *CDKN1A* expressions). EBV-positive tumors were detected by using quantitative PCR.

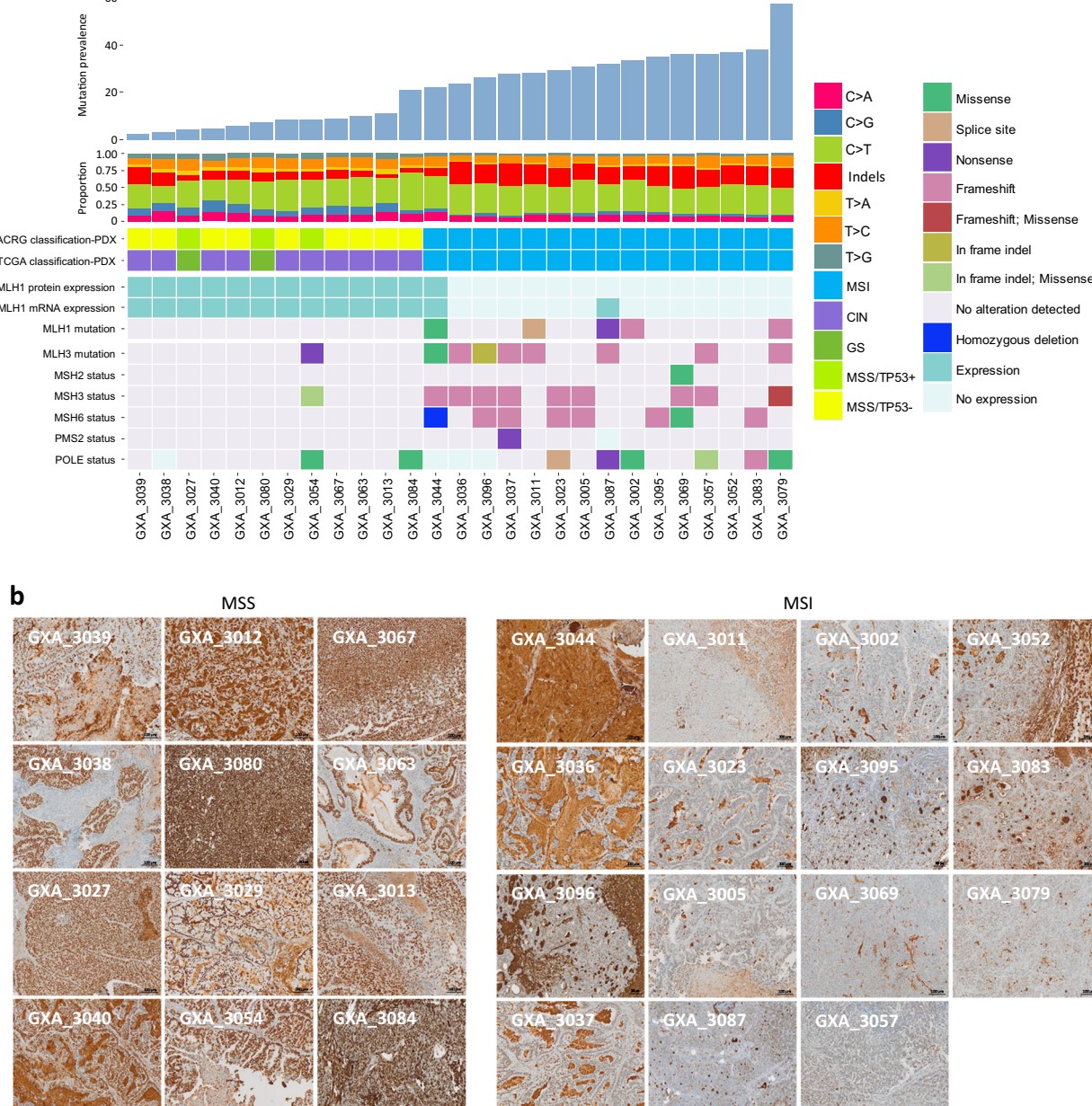

**Fig. 2 Mutational prevalence, mutational signature, and genomic rearrangement of gastric cancer PDX. a** Bar plot of PDX mutation prevalence ranked from the left to the right by increasing mutation counts per megabase. Histogram of proportions of nucleotide transversions, transitions, and indels in the corresponding PDX with indication of the molecular subtypes following the TCGA and ACRG classification. mRNA expression and mutation status of *MLH1* (qRT-PCR data and whole exome sequencing data, respectively) and others mismatch repair genes (*MSH2, MSH3, MSH6, PMS2, POLE*; for these genes loss of mRNA expression was determined by Affymetrix HGU133 Plus 2.0 analysis) in the 27 PDX with indication of the mutation types. **b** MLH1 protein detection performed by immunohistochemistry for the 27 PDX. Tumors were categorized as MSI in case of absence of a homogeneous MLH1 chromatin staining.

In contrast, the 12 PDX models derived from MSS tumors were characterized by a high number of genomic rearrangements, a higher proportions of C > G and C > A transversions and a mutational signature associated with APOBEC hyper-activity (Fig. 2, Supplementary Fig. 2 and Supplementary Data 7). MSS/TP53[−] and MSS/TP53[+] PDX were differentially associated to signatures 2 and 13 (associated with APOBEC hyperactivity), 11 (associated to temozolomide sensitivity), 14, 18, 19, and 20. The Affymetrix SNP6.0 microarray analysis revealed that MSS PDX showed higher ploidy in contrast to MSI PDX (mean$_{MSI\ ploidy}$ = 2.29, mean$_{MSS\ ploidy}$ = 3.09) (Fig. 3a and b).

Detailed analysis of somatic copy number alteration (SCNA) allowed to distinguish 10 CIN PDX with high SCNA values (from 42 to 258, mean = 115) and two GS PDX with lower SCNA values (SCNA = 13 and 15) among the 12 MSS PDX (Fig. 3a and c and Supplementary Data 8). The genomic identification of significant targets in cancer analyses (GISTIC) led to identify recurrent focal deletions (3p14.2 (*FHIT*), 11q11, and 4q13.2) and focal amplifications (17q13 (*ERBB2*), 8q24.21 (*MYC*), 3q26.31 (*PIK3CA, TRAIL*), 8q24.22, 8q21.13) (Fig. 3d and Supplementary Data 9). In contrast, only few focal rearrangements were observed in MSI PDX (Fig. 3d and Supplementary Data 10).

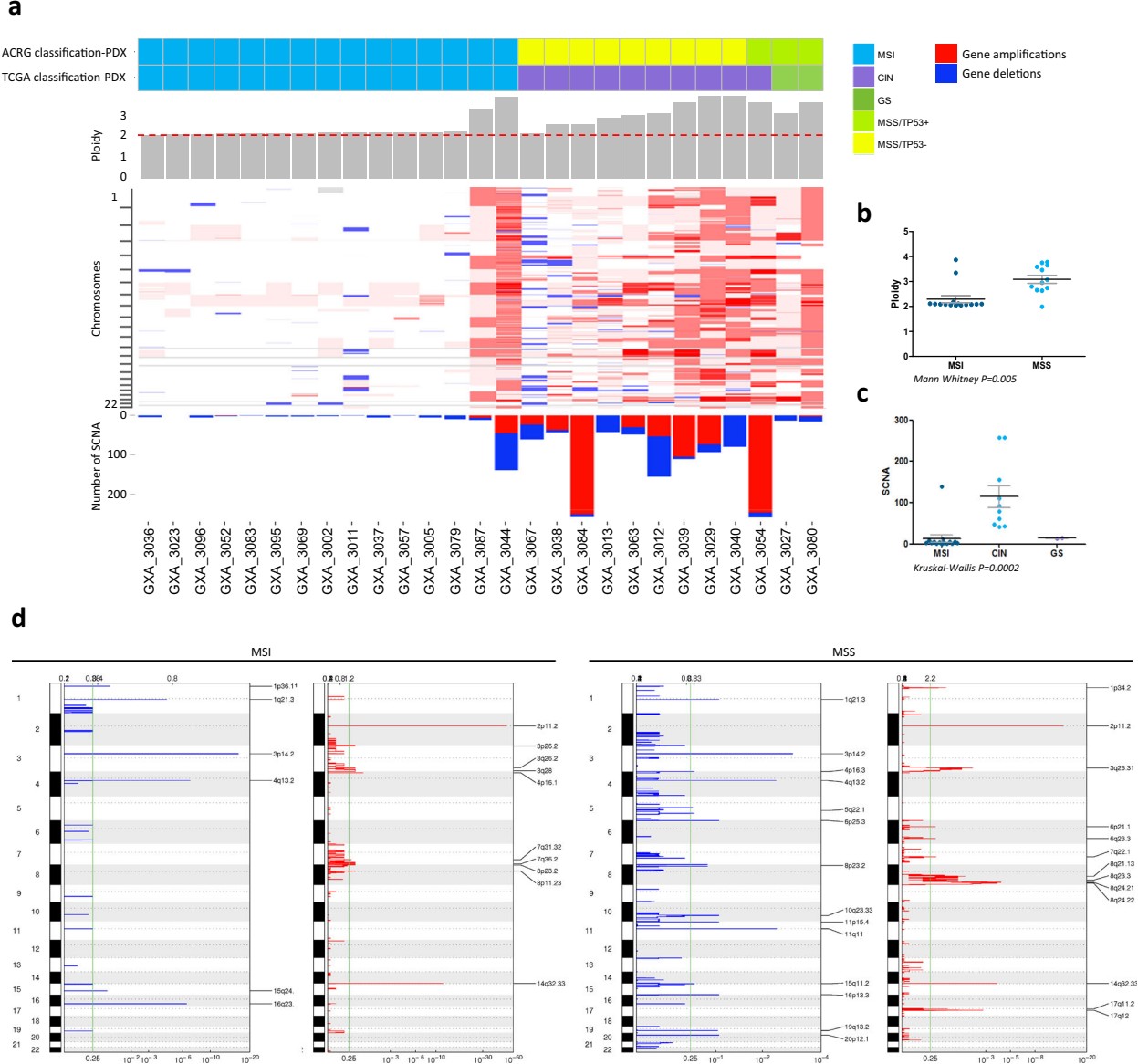

**Fig. 3 Chromosomal aberrations in gastric cancer PDX. a** PDX were sorted by molecular subtypes and increasing ploidy. The molecular subtypes following the TCGA and ACRG classifications are indicated above a gray bar plot indicating the overall ploidy of each PDX. Heatmap of gene copy number alterations according PICNIC analysis. Bar plot showing the counts of somatic copy number alterations (SCNA, homozygous deletions (PICNIC = 0) and amplifications (PICNIC ≥ 8)) per PDX. **b** Ploidy compared between the PDX of the MSI ($n = 15$) and of the MSS ($n = 12$) subtypes (Mann–Whitney test). The black bars represent the mean and the gray bars the standard error of the mean. **c** Dot plot comparing the number of SCNA between the molecular subtypes (15 MSI, 10 CIN, and two GS PDX, Kruskal–Wallis test). The black bars represent the mean and the gray bars the standard error of the mean. **d** GISTIC 2.0 analyses of focal amplifications and deletions in the MSI and in the MSS gastric PDX. Chromosomal locations of peaks of significantly recurrent focal amplifications (red plots) and deletions (blue plots) are plotted by false discovery rates. Peaks annotated by cytoband have a false discovery rate <0.25 (green line).

At transcriptome level, we observed importantly that 89% of PDX retained the gene expression signature determining ACRG subtypes of parental tumors (Supplementary Data 6, Chi-square test $p = 0.0003$). Briefly, $n = 9$ PDX were classified as MSS/TP53−, while $n = 3$ were MSS/TP53+ that includes the two GS.

Finally, by investigating 48 genes with potentially targetable alterations identified in PDX models using a database of genomic biomarkers for cancer drugs and clinical targetability in solid tumors[20], we observed that the collection had very similar distribution of these alterations than those observed across TCGA tumors.

MSS/TP53+ PDX harbored the lowest number of potentially targetable alterations compared to MSI PDX. Overall, *TP53* and *MSH3* mutations were the most frequent alterations (63% and 52%) in both MSS and MSI PDX and may allow investigation of compounds such as *WEE1* and DNA-PKcs inhibitors (Fig. 4a). In the MSI PDX, alterations in genes such as *ATM*, *MSH3*, *BRCA1*, and *BRCA2*, suggest the testing of DNA-PKcs or PARP inhibitors. Several other genes with potentially targetable alterations such as *KRAS* and *MYC* or deletion of *CDKN2A*, *CDKN2B*, suggested the investigation of MAPK pathway inhibitors, BET and PIM inhibitors as well as CDK4/CDK6 blockers. We also identified five PDX with *ERBB2* gene

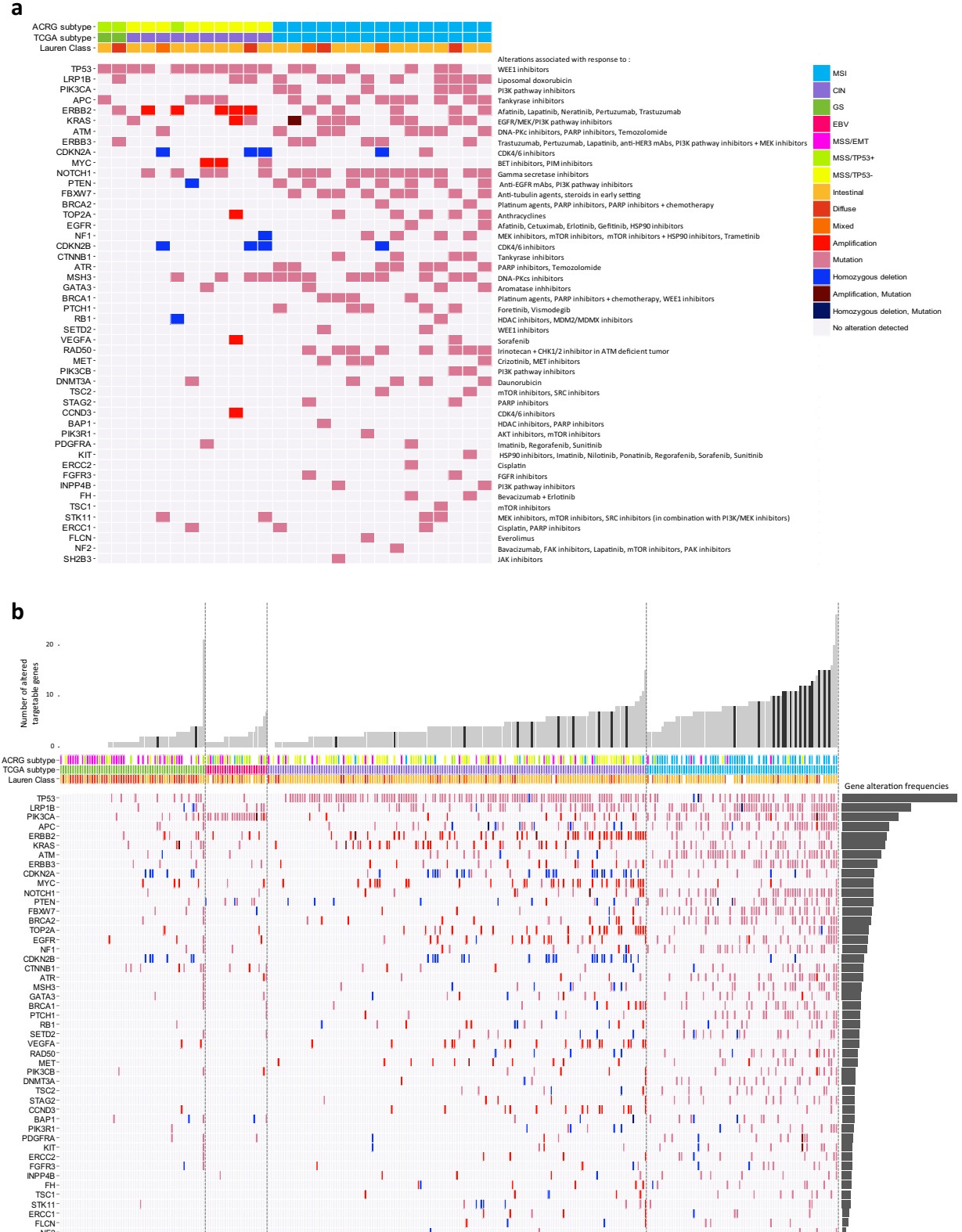

**Fig. 4 Gastric cancer PDX landscape of potentially targetable alterations and comparison with gastric cancer patient tumors from the TCGA. a** Landscape of potentially targetable gene alterations and corresponding therapies in PDX. Models were investigated for potentially targetable alterations and corresponding putative sensitivity towards various therapies as reported in https://www.synapse.org/#!Synapse:syn2370773 database. PDX were ranked from the left to right by subtypes as determined by the ACRG and TCGA classifications and by increasing number of altered genes. **b** Landscape of the potentially targetable alterations in the 27 PDX was merged with those of the patient tumors from the TCGA (n = 295). Bar plot showing the number of altered genes per sample. The samples were classified according the TCGA subtypes and by increasing number of altered genes. Gray bars were patient tumors and black bars represented PDX samples. Below are indicated the ACRG and TCGA subtypes and the Lauren subtypes. Horizontal bar plot showed the gene alteration frequencies observed throughout the PDX and tumor samples.

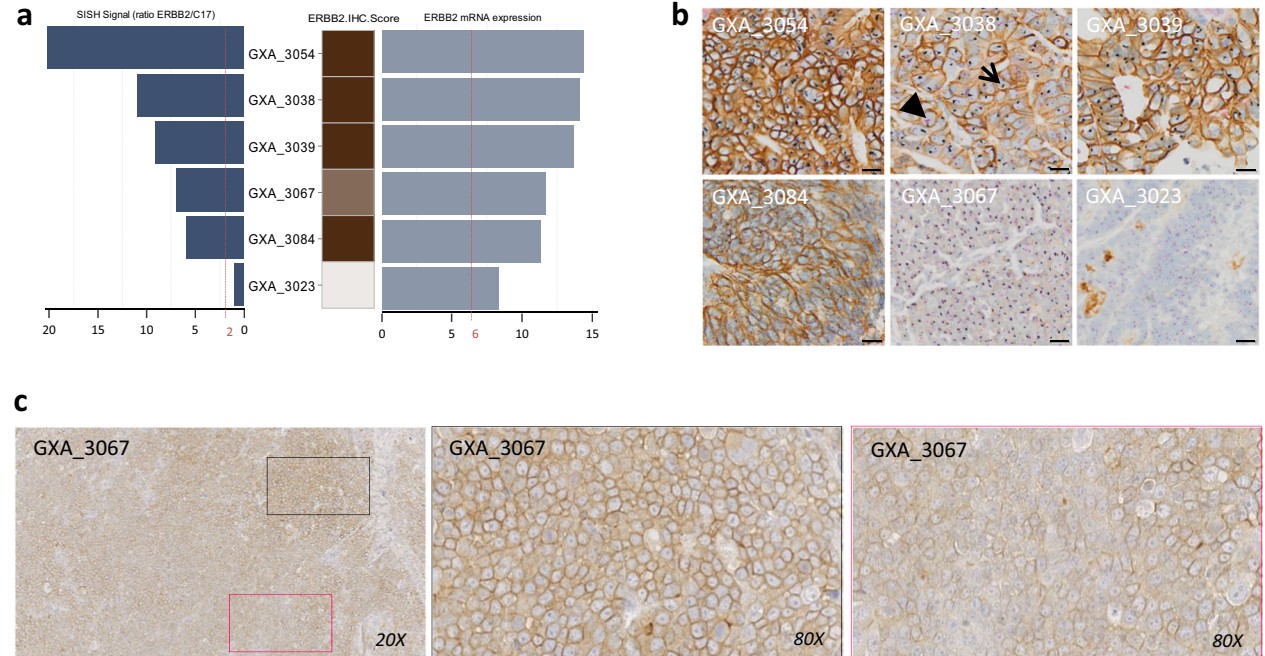

**Fig. 5 Levels of gene amplification and overexpression in ERBB2-amplified gastric cancer PDX. a** The mRNA expression is given by the Affymetrix probe 216836_at (Log2 values) and the gene copy number by silver in situ hybridization ratio *ERBB2/cep17*, the ERBB2 immunohistochemistry scores of the corresponding PDX are indicated by a white to brown color code on the right of the graphic. The red dashed lines represent the threshold of expression for Affymetrix values (6) and the threshold of amplification as determined by silver in situ hybridization (2). **b** Representative images of detection of *ERBB2* amplification and overexpression by double silver in situ hybridization and immunohistochemistry staining in five *ERBB2* amplified models (GXA_3054, GXA_3038, GXA_3039, GXA_3084, and GXA_3067) and in one non-*ERBB2*-amplified model (GXA_3023) for comparison at ×20 magnification. Scale bars represent 50 μm. The arrow indicates the ERBB2 staining (blue) and the arrow head indicates the cep17 staining (pink). **c** Representative image of ERBB2 heterogeneous signal observed in GXA_3067 (score 2, homogeneity 70%) at ×20 (left) and ×80 magnification (middle and right).

amplification (Fig. 5a and b) with *n* = 4 PDXs showing mRNA and protein overexpressions (IHC score 3) and one PDX (GXA_3067) showing heterogeneous protein expression (IHC scored 2+, 70% of positive cells) (Fig. 5c).

For comparison, we analyzed the distribution of these gene alterations in both the 27 PDX and the 295 patient tumors from the TCGA (Fig. 4b). The overall number of altered genes per sample was comparable in patient tumors and PDX models, and was as expected, dependent on the histological and molecular subtypes (bar plot). However, the PDX are frequently classified among the heavily altered samples. At individual gene levels, the percentages of gene alteration observed per subtype correlate between patient tumors and PDX models (Spearman correlation *r* = 0.67 and 0.55 for MSI and CIN, respectively, Supplementary Data 11, Supplementary Fig. 3a and b). However, differences were noticed. A higher proportion of alterations in *NOTCH1, MSH3*, and *RAD50* genes was present in MSI PDX compared to MSI patient tumors (80%, 60%, and 54% in MSI PDX, and 29%, 11%, and 13% in the MSI patients, respectively), while the percentage of alterations in BRCA2 was higher in MSI patient tumors than PDX models. Similarly, for the CIN subtype, PDX were enriched in *NOTCH1, MSH3*, and *ERBB2* gene alterations compared to patient tumors. Interestingly, in MSS tumors, samples heavily affected by gene deletions/amplifications were also frequently accompanied with highest mutation loads in the 48 genes with potentially targetable alterations (Spearman correlation *r* = 0.81 and 0.73 for the MSS of the TCGA and the ACRG classifications, respectively; see Supplementary Fig. 3c and d and Supplementary Data 12).

We also observed that the models with higher number of alterations in these 48 genes had shortest growing time at P1 (Spearman correlation *r* = −0.44, *p* = 0.028). Also, the time

between surgery and first implantation exceeding 5 days negatively impacted establishment of MSS subtype (Mann–Whitney *p* = 0.0081).

**Clonal selection and evolution during PDX establishment**. In our study, eight PDX models (four MSI and four MSS) had sufficient DNA available from tumor (T), matched normal (N), first three PDX passages (P1–P3, one tumor sample analyzed per passage for a given PDX model) and the established PDX (after P4, one tumor sample analyzed per PDX model) to attempt genetic analysis of the established models in context of parental tumors and early passages. As expected, the whole exome sequencing analysis (see Supplementary Data 13 for technical details) confirmed that MSI tumors and corresponding PDX were hypermutated and had a different mutational signature compared to MSS tumors (Fig. 6a and b). We observed that MSS PDX had stable mutation loads across passages, while MSI samples presented a trend of increased mutation burden, through indel increase in the passages.

To study larger variations at chromosome levels, we investigated the allelic fractions of all single nucleotide variants (single nucleotide polymorphisms, germline, and somatic mutations). The MSI tumors were more likely to retain chromosome stability across PDX establishment with the allelic fractions close to 0, 0.5 for the majority of the chromosomes, in contrast with MSS PDX that showed more chromosomal instability (Fig. 6c). In MSI, a notable exception of chromosome 7 was however seen. All passages in all tumors showed a copy number increase (green arrowhead). We noticed that the MSI GXA_3037 series underwent dramatic changes with large numbers of tumor-specific variants lost in P1 along with a distinct set of somatic variants

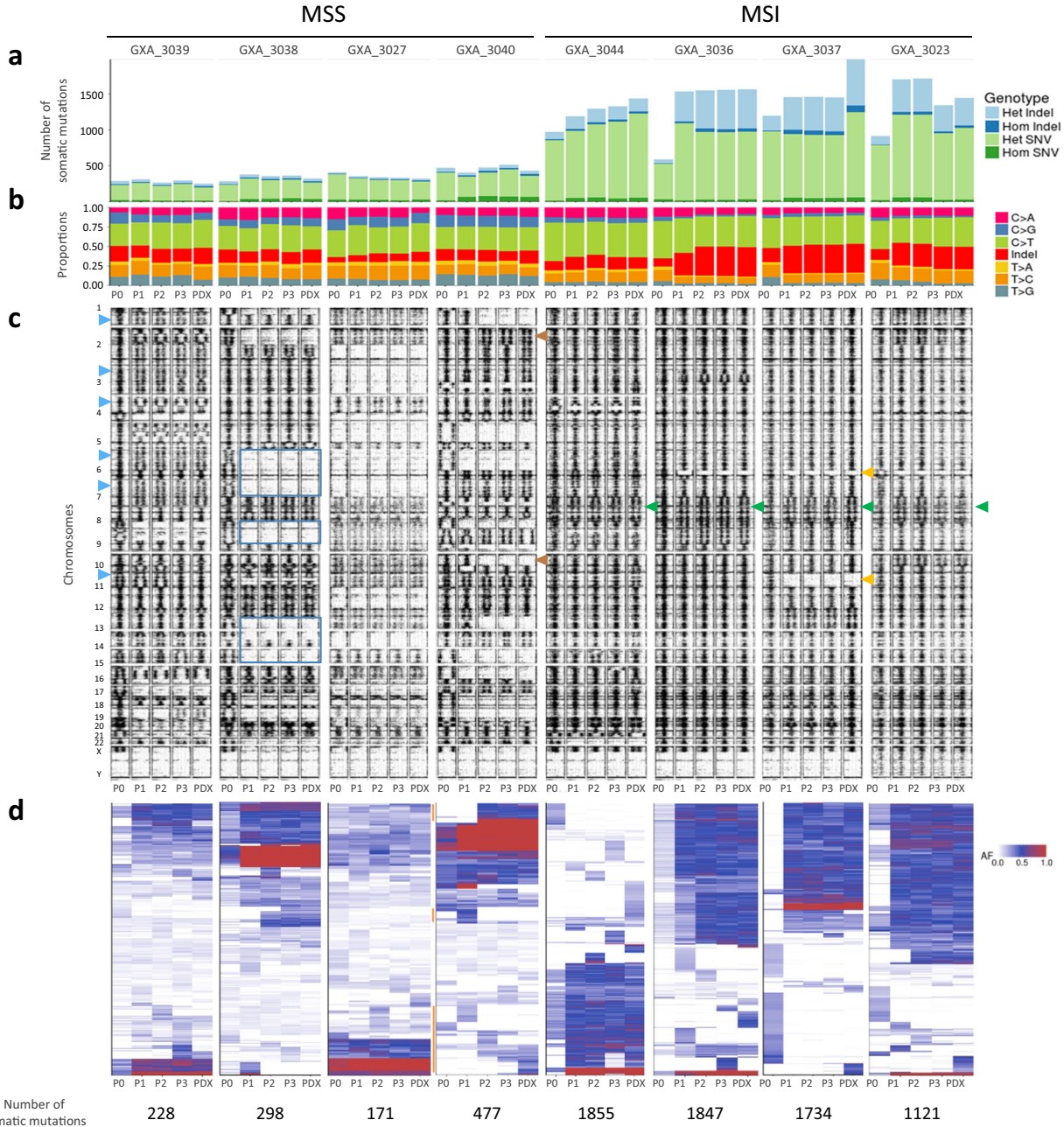

**Fig. 6 Gastric cancer PDX clonal variation through passages. a** Bar plot showing the number of somatic mutations in patient tumors (P0), PDX at passage 1, 2, and 3 (P1–P3) and the corresponding established PDX (indicated as "PDX"). Indels are shown in blue and small nucleotide variants in green. Heterozygous mutations are indicated in light blue and light green, while homozygous mutations are in dark blue and dark green. **b** Histogram of proportions of nucleotide transversions, transitions and indels. **c** Allelic read frequency (AF) of variants detected by whole exome sequencing. Each point represents a genomic single-nucleotide variation (polymorphism or mutation). AF is shown per passage on the *x*-axis ranging from 0 (left) to 1. Labels on the *y*-axis show the start of the individual chromosomes. For genomic regions with two alleles, the AF is expected to be close to 0, 0.5, or 1 while aberrations from this pattern hint toward loss of heterozygosity (no 0.5 AF) or copy number increase (more than three bands, e.g., at 0, 0.33, 0.66, and 1 for three copies). **d** Hierarchical clustering of somatic mutations (indels and small nucleotide variants) identified at P0, 1, 2, 3 and in the established PDX, by using whole exome sequencing with a minimum of 10 read coverage in all samples of the same model.

appearing in P1, which then remained consistent in P2, P3 and in the established PDX model (examples of losses of heterozygosity indicated by orange arrow heads). In contrast to MSI, multiple aberrations hint toward loss of heterozygosity or copy number increase were seen in MSS samples. As example losses of heterozygosity for chromosome 5, 6, 8, 12, 13, and 14 in P1–P3 of GXA_3038 (dark blue rectangles), copy gain in most chromosomes of GXA_3039 (light blue arrow heads). In GXA_3040 we

observed a loss of heterozygosity on the distal end of chromosome 1 and 9 between the original tumor and P1 (brown arrow heads), an expansion of this loss of heterozygosity between P1 and P2 that stayed stable between P2 and P3, suggestive of selection happening during passages. For GXA_3027, GXA_3039, and GXA_3040 the findings were validated by using data from Affymetrix genome wide SNP6.0 assay and ASCAT[21] algorithm (Supplementary Fig. 4).

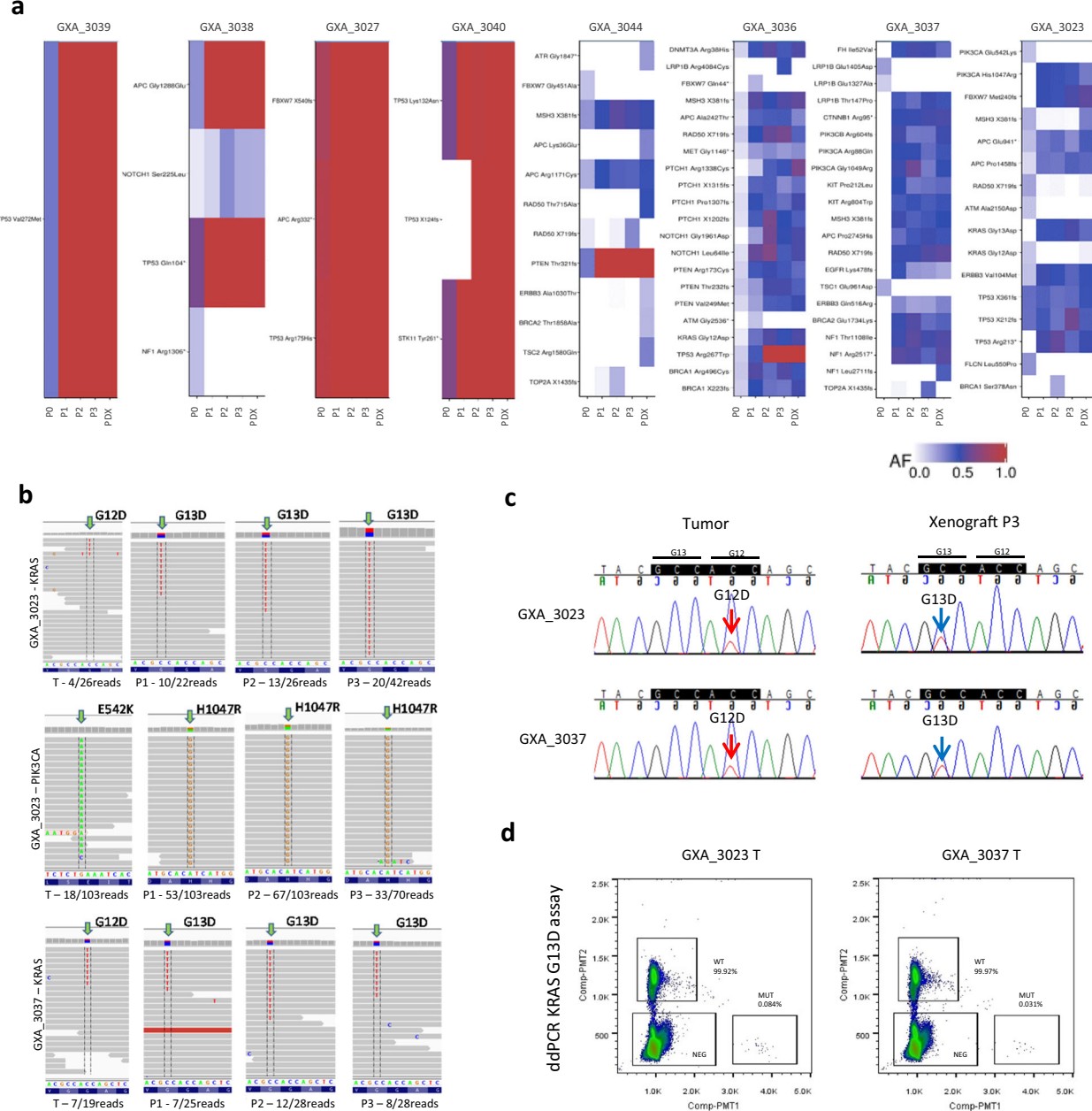

**Fig. 7 Mutations in potentially targetable genes across passages. a** Allelic read fractions (AF) for the 48 potentially targetable genes in tumors and corresponding PDX across passages. **b** Whole exome sequencing read pileup in GXA_3023 showing *KRAS* p.G12D and *PIK3CA* p.E542K in tumor (T) but *KRAS* p.G13D and *PIK3CA* p.H1047R in P1–P3. Also, in GXA_3037, *KRAS* p.G12D in tumor, p.G13D in P1–P3. **c** Sanger sequencing confirming the *KRAS* mutations observed by whole exome sequencing. **d** Droplet digital PCR confirming p.G13D exists as minor clonal in the tumors of GXA_3023 and GXA_3037.

We studied the single nucleotide variants and small indels with a minimum of 10 read coverage across tumors and PDX to evaluate the representativeness of the PDX models regarding the mutation pattern observed in the patient tumors. The mutation contents and their allelic fractions overall were more stable in MSS series than in MSI samples (Fig. 6d, Supplementary Data 14). Exception was for the MSS model GXA_3040, with some mutations appearing and disappearing from P1 to the established PDX (see orange bars on the left of the plot). In all MSS samples, we noticed a trend for an increase in allelic fraction of some mutations starting at P1, that can be due to the replacement of the human stroma by the mouse stroma. A higher variability of mutational profiles was seen during MSI model establishment.

Three subclasses of mutations were identified, those stable over passages having usually a high allelic fraction; those presenting an increase of their allelic fraction at P1, probably due to a clonal selection in the passages (e.g. GXA_3037, at passage 1) and/or removal of human stromal cells (e.g. GXA_3040), and those with a low allelic fraction that appeared and disappeared over passages most likely being a consequence of the clone composition of the tumors over passages.

We explored gain/loss of variants in greater detail by focusing on allelic read fractions of 48 potentially targetable cancer genes identified within the PDX collection[20] (Fig. 7a and Supplementary Data 15). We observed only few changes of the cancer genes variants across passages in MSS samples. It corresponded mostly

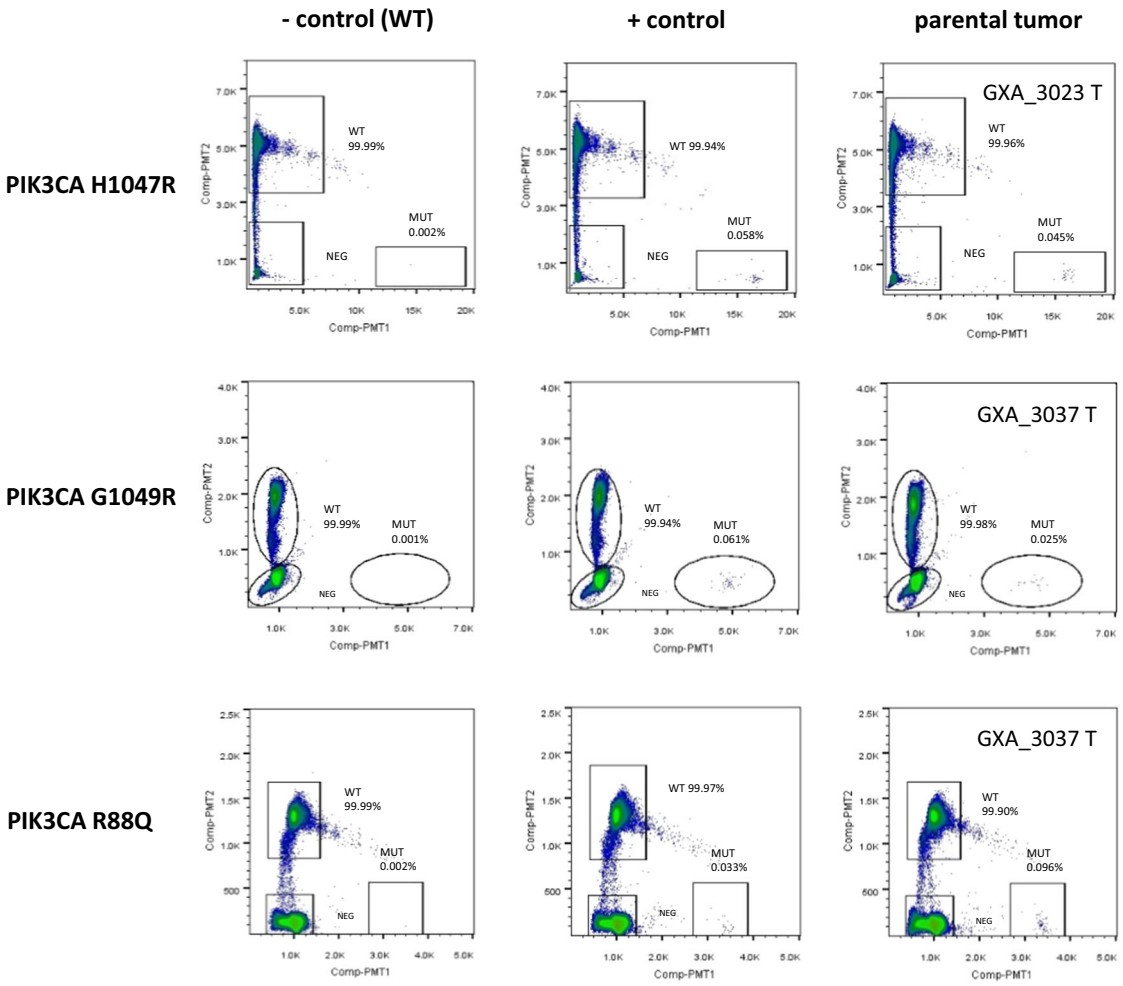

**Fig. 8 RainDrop ddPCR of three *PIK3CA* assays also detected minor clones in parental tumors that were enriched in xenograft passages.** *PIK3CA* H1047R mutation in GXA_3023 tumor, and *PIK3CA* G1049R, R88Q in GXA_3037 tumor are detected at very low percentage, close to the 0.1% mutant allele frequency in the positive controls. The results confirmed that the *PIK3CA* mutants, consistently detected in all the passages, originated from minor clones in the parental tumors.

to variants detectable at P0 for which the allelic read fractions increased to 1 (homozygous somatic mutations) at P1 and remained stable after. Only a *NOTCH1* variant presented a slight allelic fraction increase over passages in GXA_3038 and one of the two *TP53* variants in GXA_3040 that were not detectable at P0, appeared at P1. In addition, only the *NF1* variant found in GXA_3038 disappeared after P0. In contrast, more variations in allelic read fractions were seen in MSI samples. The variations affected cancer genes such as *BRCA1*, *TP53*, *KRAS*, and *PIK3CA*. These variants were frequently not detectable at P0 and became detectable at P1 with the allelic fractions centered on 0.5 (heterozygous somatic mutations) in models at P1–P3 and in the established PDX at later passages. We also noticed a decrease of the allelic fraction for some mutations that were detectable at P0 and not detectable in PDX. It argued in favor of a clonal selection occurring mainly during P0 to P1. Indeed, we noticed in the GXA_3023 a somatic mutation *KRAS*$^{G12D}$ and *PIK3CA*$^{E542K}$ in the parental tumor but the presence of *KRAS*$^{G13D}$ and *PIK3CA*$^{H1047R}$ across the passages. Similarly, we observed in GXA_3037 a somatic mutation *KRAS*$^{G12D}$ was lost and replaced by a *KRAS*$^{G13D}$ over passages. These two samples had shown evidence of a clonal selection across passages (Fig. 7b and c).

To ascertain if the two PDX models acquired these mutations as a de novo event or a rare subset of cells was selected during first passage, we deployed a highly sensitive technique—droplet digital PCR. We observed (Fig. 7d) very low percentage of cells with *KRAS*$^{G13D}$ in both GXA_3023 and GXA_3037 P0, strongly suggesting the presence of rare cells that were selected during early passage. We also observed a similar situation for *PIK3CA* (Fig. 8), overall suggesting that rare cells were likely selected during PDX establishment. The selection pressure favoring expansion of cells with *KRAS*$^{G13D}$ over *KRAS*$^{G12D}$ clone in the passage establishment needs to be further investigated.

## Discussion

Over the last decade, propagation of patient tumors in PDX is increasingly being used as a model system in anti-cancer drug discovery and development as well as for biomarker investigation[22,23]. It is therefore important to understand the intra-tumor and inter-tumor heterogeneity that exists in both the parental tumors and the established PDX models, so that PDX models can be optimally utilized. To this end, we developed a collection of PDX from gastric cancer tumors and investigated in detail their clinical and molecular patterns. We show that the PDX establishment success largely relies on both tumor histological and molecular subtypes. We also observed that PDX are subject to clonal selection in early passages.

Firstly, we observed that tumors of the Lauren's intestinal subtype were established more commonly than diffuse or mixed tumors, confirming previously published data[9,10,12]. Secondly, the

analysis revealed that not all gastric cancer molecular subtypes were established, with PDX predominantly developed from MSI, CIN, and MSS/TP53⁻. In contrast, PDX were rarely or not developed from the molecular subtypes EBV, GS, and MSS/EMT or MSS/TP53⁺ tumors; based on TCGA and ACRG studies, respectively. The MSI tumors accumulate a high number of mutations. This characteristic may confer a certain adaptability to the tumor cells and thus a facility to grow in a new micro-environment (the immune-compromised mice). PDX were also frequently established from ERBB2-positive tumors, probably because of the capacity of these cells to proliferate without the expression of the corresponding ligand (ligand-independent growth). Other subtypes, such as the GS, may require additional growth factors to proliferate which might not be available in the immune-compromised mice.

This study has important implications. Firstly, the commonly held belief that PDX reflect parental tumors, needs to be adjusted in the context of this data and other recently emerging data[24] suggestive of inadequacies in PDX models. Both, establishment bias and clonal selection during PDX establishment happen making these models differ both at gastric cancer population level, as well as at the level of parental tumors. This study demonstrates that the consideration of gastric cancer PDX models on the simple basis of their type or histology is not sufficient. Molecular characterization, in terms of gene mutations, gene expression, and gene copy number, may drive appropriate use of these PDX for drug testing experiments. Secondly, in the field of biomarker discovery from PDX for gastric cancer treatments, molecular subtypes existing in PDX is likely an important consideration at the time of translating findings from preclinical to clinical settings. A careful selection of PDX models based on their molecular pattern may increase the success of drug testing experiments and/or may allow identifying molecular determinants of the sensitivity response. Thirdly, Avatar and co-clinical trials have been discussed and are being implemented[1,25] in the clinical trial NCT02732860: "Personalized Patient-Derived Xenograft (pPDX) Modeling to Test Drug Response in Matching Host (REFLECT)". Our results highlight clonal selection events which can occur during early PDX establishment with the emergence of rare clones. This may have implications for results and interpretation of data avatar trials.

Our study has a few known limitations. First, it is possible that the selection of tumor fragments for establishment and characterization, degree of immuno-deficiency in the mouse strain (for e.g. SCID vs. NSG) and the environmental context (lack of certain growth factors) may influence the establishment rate as well as clonal selection. Recent work by Eirew et al.[26], however, argues against it, who observed similar phenomenon while using NSG and NRG mice in establishing breast cancer PDX. Secondly, our analysis was conducted on bulk tissue samples. With the advent of newer methodologies such as single cell sequencing, a more comprehensive picture of heterogeneity in tumors and PDX may emerge. Thirdly, surgical samples may not capture the heterogeneity of metastatic samples and may have differing dynamics with regards to establishment rates and clonal selection. Fourthly, our study is focused on one tumor type. However, similar findings have been observed in cell lines and PDX[17,24,26–28], likely suggesting the existence of such phenomenon and an important consideration in other tumor types and associated tumor model system.

In summary, we showed this gastric cancer PDX collection does not fully cover the diversity of gastric cancers. Within the established models, molecular subtypes and possible clonal evolution raises the possibility of this being an important consideration for various translational studies. We also provide a molecular investigation framework that may aid in rational use of PDX models for translational studies not only in gastric cancer but also in other tumor types as well.

## Methods

**Study design, patient tissue specimens, and pathology.** We designed this study as a patient tumor-derived xenograft study in gastric cancer with no pre-specified hypothesis. We systematically collected n = 100 surgical tumors from a single institution, i.e. Seoul National University at the time of total or sub-total gastrectomy. We aimed to establish a collection of Asian gastric cancer PDX as well as understand histological, genetic, and other clinic-pathological biases seen during PDX establishment. We clinically annotated the tumors but de-linked them from personally identifiable information. All patients provided informed consent and SNU IRB approved the study (IRB number H-0807-037-250).

Final cohort comprised of n = 64 males and n = 36 females, with patient ages ranging from 37 to 90 years (median = 64 years). Of the n = 100 patients, n = 10 patients showed metastasis at the time of surgery while no evidence of metastasis was found in n = 88 patients (data not available for two patients). According to the Lauren classification[15], n = 53/100 tumors were intestinal, n = 42 diffuse, and n = 5 mixed (see details in Table 1).

We received patient material at former Oncotest GmbH now Charles River Discovery Research Services GmbH from Seoul National University, College of Medicine, 24–72 h after surgery. Tumor materials were collected under sterile conditions directly after surgery, for PDX model establishment and molecular profiling. We stored one piece of tumor (~1 cm³) in Aqix (Liquid Life) medium for further implantation in nude mice. Additionally, we snap froze in liquid nitrogen one piece of the tumor as well as a fragment of normal peritumoral tissue (~1.5–2 cm³ for each piece) and stored it at −80 °C for DNA and RNA extractions. Finally, a third piece was directly fixed with 5% formalin for 24 h for FFPE blocks preparation for clinical investigation.

**MSI analysis.** MSI analysis was performed as previously described[29]. Briefly, MSI status was determined by analyzing five microsatellite loci (BAT-26, BAT-25, D5S346, D17S250, and S2S123) using DNA auto-sequencer (ABI 3731 genetic analyzer; Applied Biosystems, Foster City, CA). According to the Bethesda guideline, tumors were classified as MSI-H when at least two of the five markers displayed additional bands compared to the corresponding normal tissue, MSI-L, when additional alleles were observed with one of the five markers, and MSS, when all microsatellite markers examined displayed identical patterns in both tumor and normal tissues. MSI-H tumors were classified as "MSI" and MSI-L or MSS samples were categorized as "MSS".

**PDX model establishment and animals.** Female NMRI nude mice were obtained from Harlan (Denmark) at age of 4–6 weeks. Pieces of ~1–2 mm³ of tumors were implanted on these immune-compromised mice. This study was carried out in strict accordance with the recommendations in the Guide for the Care and Use of Laboratory Animals of the Society of Laboratory Animals (GV SOLAS). All animal experiments were approved by the Committee on the Ethics of Animal Experiments of the regional council (Regierungspräsidium Freiburg, Abt. Landwirtschaft, Ländlicher Raum, Veterinär- und Lebensmittelwesen—Ref. 35, permit-#: G-13/13).

**Immunohistochemistry (IHC) and in situ hybridization analyses.** MLH1 IHC analysis was performed on a Ventana benchmark autostainer with the Ventana MLH1 antibody (clone M1, Cat. No. 790-4535), according to the manufacturer's instructions. PDX were categorized as MLH1 negative in case of absence of a homogeneous staining. ERBB2 IHC and silver in situ hybridization (SISH) analyses were performed on a Ventana Benchmark autostainer, using the Ventana anti-HER2/neu antibody (Cat. No. 790-2991) and the Ventana INFORM HER2 Dual ISH DNA Probe Cocktail (Cat. No. 780-4422), respectively, according to the manufacturer's instructions. Evaluation of ERBB2 IHC and SISH status was performed according to the FDA guidelines.

**DNA and RNA samples preparation.** DNA and total RNA were extracted from frozen patient tumors and PDX material as previously described[30]. In brief, DNA was extracted from snap frozen patient tumors or PDX. A piece of ~40 mg of frozen tumor was cut per sample and digested with proteinase K buffer (Qiagen, Hilden, Germany) overnight at 55 °C, followed by a DNase-free RNase digestion (Qiagen, Hilden, Germany). The DNA was subsequently extracted with phenol/chloroform/isoamyl alcohol, precipitated and washed with ethanol, and resuspended in Tris–EDTA buffer (Tris 10 mM pH 8, EDTA 0.1 mM pH 8). The DNA integrity of each preparation was checked on a 1.3% agarose gel, and the purity analyzed with a NanoDrop 2000 spectrophotometer (Thermo Fisher Scientific, Waltham, MA, USA).

For RNA preparation, a piece of ~40 mg of frozen tumor was cut per patient sample and used for the extraction, while four pieces of ~40 mg were pooled per PDX to limit the inter/intra-tumor variability. These pieces of frozen tissues were used as starting material for the RNA extraction using the mirVana™ miRNA isolation kit (Ambion, Carlsbad, CA, USA) according to the manufacturer's instructions. The RNA quality was controlled for purity with the NanoDrop 2000

(Thermo Scientific, Waltham, MA, US) and the RNA integrity by a Bioanalyzer (Agilent, Agilent Technologies, Palo Alto, CA, USA).

**quantitative PCR (qPCR) determination of the EBV subtype**. EBV infection load was determined in 40 cycles on a StepOnePlus™ (Applied Biosystems) qPCR assay by using 2X KAPA™ SYBR Green Fast qPCR kits (KAPA Biosystems) in conditions recommended by the manufacturer. The EBV qPCR primers are listed in the Supplementary Table 1. EBV infection load was determined by normalizing the EBNA1 Ct (cycle threshold) values to 40 (the maximum Ct value) and divided by 1000: $EBV_{load} = 2^{(40-Ct_{EBNA1})}/1000$ and expressed in arbitrary units. Samples with $EBV_{load}$ qPCR values >1000 arbitrary units were considered with infection burden and were classified as of the EBV subtype.

**quantitative real-time-PCR (qRT-PCR) analyses**. In brief 1 µg RNA was reverse transcribed into cDNA by using MMLV reverse transcriptase. The resulting cDNA were analyzed in 40 cycles on a StepOnePlus™ (Applied Biosystems) using 2X KAPA™ SYBR Green Fast qPCR kits (KAPA Biosystems) in conditions recommended by the manufacturer. The sequences of primers of the investigated genes are listed in Supplementary Table 1. qRT-PCR of the 18S ribosomal RNA was performed for final gene expression data normalization and relative quantification as follows: Gene of interest$_{exp} = 2^{(Ct_{18s}-Ct_{Gene of interest})}$, results were expressed in arbitrary units. The gene signature calculations were done as follow: TP53 activation score = $CDKN1A_{exp}$/median ($CDKN1A_{exp}$) + $MDM2_{exp}$/median ($MDM2_{exp}$); Proliferation score = $TOP2A_{exp}$/median ($TOP2A_{exp}$) + $MKI67_{exp}$/median ($MKI67_{exp}$). Samples with TP53 activation score above 2.5 arbitrary units were considered as TP53 activated. MSS samples with loss of CDH1 had CDH1 expression values below 40 arbitrary units. Proliferation subclasses were defined as: low if values were below 1 arbitrary unit, intermediate for values between 1 and 4, and high for values above 4 arbitrary units.

**Whole somatic exome mutation analysis**. DNA were prepared as previously described[30] and were profiled by whole exome sequencing. Exons from DNA samples were targeted using Agilent SureSelect Human All Exon V1 38 MB (5), V4 51 MB (20), or V5 50MB (2) kits. Enriched genomic DNA was sequenced with Illumina HiSeq-2000/2500 in 100 or 125 bp paired-end reads and an expected coverage of ~100×. Paired-end reads were independently mapped to the Human hg19 and the Mouse mm10 reference genome with Burrow–Wheeler aligner (BWA[31]) with default parameters. To remove the mouse reads from the tumor stroma, paired-end reads that mapped better on the mouse (mm10) than on the human genome (hg19) were discarded from the human mapped read dataset (based on the BWA mapping score) using PicardTools[32]. Then, this filtered human-mapping dataset was recalibrated with GATK Lite's BaseRecalibrator[33] function after duplicates removal and indel (insertion-deletion) local realignment. Reads mapped around indels were realigned using the GATK Lite's IndelRealigner function before performing the variant calling step. Variants were detected independently using three different variant callers: GATK Lite's UnifiedGenotyper, the combination of Samtools mpileup[32] and bcftools caller[34], and Freebayes[35]. Only variants identified by all three tools, showing a minimum number of variant-supporting reads of three and a minimum allelic frequency of 5% were further analyzed. Candidate mutations were identified with SnpEff[36] by selecting only small nucleotide variants and indels with a high or moderate protein impact from UCSC or Ensembl transcripts, and by filtering out known polymorphisms from annotation databases if a variant (1) has at least three allele counts from Hapmap or CGI 69 genomes or EVS+1000 genomes or (2) shows more than 5% of minor allele in at least one population from dbSNP. Raw reads were subjected to fastQC[37] to calculate read quality metrics. After the alignment to the Human reference genome and Mouse reads removal, the quality of BAM files was assessed by Qualimap[38] to obtain the percentage of mapped reads and coverage of reads to the targeted exons. Variant detection analysis was QC-evaluated by computing and validating the transition/transversion ratio from SNPs found in exons. The on-target coverage obtained ranged from 99× to 215×, with a mean of 131×. The reads obtained were aligned against the human and the mouse genomes. The percentage of reads that mapped to the human genome ranged from 78.7% to 98.2% (median = 94.7%) and the percentage of reads that mapped to the mouse genome ranged from 1.6% to 21% (median = 4.6%) (Supplementary Table 2). In the analysis, a total of 46,282 variants were identified, germ line variants were filtered out by removing variants (n = 4495) found in the analysis of eight associated normal gastric samples, giving finally a total of 32,416 somatic mutations.

**Wide chromosomal alteration analysis**. The detection of chromosomal alterations was performed by using the Affymetrix Genome-Wide Human SNP Array 6.0 following the standard protocol recommended by the manufacturer. According to Affymetrix guidelines, contrast quality control and MAPD threshold were set at the values of above 0.4 and 0.5, respectively. Copy number data were calculated using Affymetrix GTC v4.1 and PICNIC software provided by the Cancer Genome Project from the Welcome Trust Sanger Institute[39]. Gene amplifications were defined as gene having a PICNIC ≥ 8 and homozygous deletions of genes when the PICNIC = 0. GISTIC 2.0 method was used to identify significant focal copy number alterations as described previously[40,41]. For determining genomic stable

and chromosomal instable PDX, a cutoff corresponding to 15 somatic copy number aberrations (SCNA) as the sum of homozygous deletions and gene amplifications, have been chosen. PDX having lesser or equal to 15 SCNA on the autosomes were considered as genomic stable and those with more than 15 SCNA were categorized as CIN.

**Sanger-sequencing method**. Sanger sequencing was used to confirm exome-sequencing results. Primers surrounding the variant were designed with the online program Primer3, the primer sequences were: KRAS_F2: GGTGGAGTATTTG ATAGTGTATTAACC and KRAS_R2: ACCTCTATTGTTGGATCATATTCG. PCR was carried out with Advantage®2 Polymerase Mix (Clontech #639201) with Advantage 2 PCR buffer and cycled at 95 ℃ for 2 min; 35 cycles of 95 ℃ for 30 s; 58 ℃ for 30 s, 72 ℃ for 30 s, and a final extension of 72 ℃ for 10 min. PCR products were purified with Wizard® SV Gel and PCR Clean-Up System (Promega #A9281). Sequencing PCR was carried out using ABI BigDye Terminator v3.1 cycle sequencing kit (Life Technologies #4337457). The resulting products were run on an ABI 3730xl DNA analyzer. All sequences were visually analyzed with Sequencher (Gene Codes Corp.).

Sanger sequencing was also used to investigate the MSI mutation status in nine gastric patient tumors (GXA_3044, GXA_3045, GXA_3048, GXA_3050, GXA_3081, GXA_3082, GXA_3090, GXA_3092, and GXA_3094). Primers allowing complete amplification of the MLH1 coding sequence were designed using the PCRTiler v1.42tool (http://pcrtiler.alaingervais.org:8080/PCRTiler/) and are listed in the Supplementary Table 1. PCR were carried out with the KapaHiFi hot start polymerase (Peqlab #07-KK2501-02) for PCR on cDNA and Phusion DNA polymerase (New England Biolabs, # M0530L) for PCR on genomic DNA, both with high fidelity buffers, following the manufacturer's instructions. Nested PCR (30 cycles each) were done to amplify cDNA and 30 cycles-PCR were performed when genomic DNA was used as matrix. PCR products were purified using the QIAquick PCR purification kit (Qiagen, #28104) and sent to the GATC laboratory (now Eurofins Genomics, Konstanz, Germany) for Sanger sequencing.

**Droplet digital PCR method**. All RainDrop droplet digital PCR experiments were performed at RUCDR Infinite Biologics (Piscataway, NJ). Briefly, 0.1% mutant allele frequency positive controls were prepared by serial dilution of mutation-specific cell line with wild type genomic DNA (Promega), the wild type genomic DNA is also used as negative control (0% mutant). Tumors, positive and negative controls genomic DNA were sheared to ~3000 bp using Covaris focused ultra-sonicator. For each of the four mutation assays, 100 ng sheared DNA was mix with assay-specific 40X primers and probes, 2X Taqman genotyping master mix (Life Tech), 25X droplet stabilizer (RainDrop), and distilled water in 25 µl total volume. Primers and fluorescent probes used in this experiment are listed in Supplementary Table 3. Droplets containing sheared DNA and PCR reaction components were generated in RainDrop source instrument and amplified in a thermal cycler with the following cycling parameters: 10 min 95 ℃, then 45 cycles of 95 ℃ for 15 s, and 60 ℃ for 1 min, followed by 98 ℃ for 10 min. After PCR completion, droplets fluorescence was measured with RainDrop droplet reader and processed into two-dimensional scatter plot display. Appropriate gates were drawn for each droplet cluster and the number of droplets within each gate was counted.

**Gene expression profiling**. Total RNA was submitted to service providers for microarray analyses by using Affymetrix HGU133 plus 2.0 arrays. First-strand and second-strand synthesis, biotin labeling, fragmentation, and hybridization were performed according to Affymetrix protocols. Evaluation and normalization of Affymetrix GeneChip data were done in the "R" (version 2.15.3) statistical computing environment. The hybridizations were normalized by using the gc robust multichip averaging (gcRMA) method from Bioconductor to obtain summary expression values for each probe set. Gene expression levels were analyzed on a logarithmic scale.

**Statistics and reproducibility**. All the statistical tests were done in GraphPad Prism 5. Chi-square and Fisher's test were used to evaluate the association between the clinical data (gender, grade, metastatic status, differentiation, Lauren classification, vascularization, and stroma content), the mutation status and the PDX establishment success rate or the molecular groups of gastric PDX. Mann–Whitney test or Kruskal–Wallis test were used to compare the groups of gastric PDX to the clinical data (age, delay of engraftment), the number of somatic mutations, the ploidy, and to the total number of gene amplifications and gene deletions. Spearman correlation was performed to evaluate the correlation between the mean signature of mutational process of the PDX and the signatures published by Alexandrov et al.[19], to compare the percentage of alteration in the 48 genes in the patient tumors and the corresponding PDX by molecular subtype, and to compare the alteration counts to the number of copy number variations in the 48 genes with potentially targetable genomic alterations in the MSS TCGA patient tumor samples according to the TCGA and the ACRG classification.

**Reporting summary**. Further information on research design is available in the Nature Research Reporting Summary linked to this article.

## Data availability

The molecular data of the 27 established Asian gastric PDX can be queried on the Charles River Tumor Model Compendium at "https://compendium.criver.com". The whole exome sequencing data (raw FASTQ files) of the PDX models has been deposited in Sequence Read Archive (SRA) under the accession code SRP150675. The raw (CEL files) Affymetrix HGU133 Plus 2.0 transcriptomic data of the 27 Asian gastric PDX models that support the results presented in this paper has been deposited in Gene Expression Omnibus (GEO) under the accession code GSE115637. The raw (CEL files) Affymetrix SNP6.0 data and the PICNIC processed genomic data presented in this study for the 27 established Asian gastric PDX models, 7 normal tissues, 7 patient tumors, and 21 PDX samples at passages 1, 2, and 3 have been deposited in GEO under the accession code GSE115674. All Affymetrix data can be accessed via the GEO code GSE115755. The molecular data of the 295 patient tumors from the TCGA cohort (TCGA Nature, 2014, https://doi.org/10.1038/nature13480) and the associated clinical data are accessible from the cBioPortal (http://www.cbioportal.org/study/summary?id=stad_tcga_pub). The ACRG subtypes of the 295 gastric tumors from the TCGA dataset were presented in the paper published by Cristescu et al. (Nature Medicine, 2015, https://doi.org/10.1038/nm.3850) and are available upon request to Amit Aggarwal (aggarwal_amit@lilly.com). The PDX samples are the proprietary of Charles River Discovery Research Services GmbH, Freiburg Germany. The established PDX models can be used for research projects on a fee-for service model. The DNA and RNA samples prepared from the established PDX models are the proprietary of Charles River Discovery Research Services GmbH, Freiburg Germany and can be purchased on demand.

## Code availability

The statistical analyses were performed in R (version 2.15.3) and GraphPad Prism (version 5). The codes used for the genomic analysis are available upon request to the corresponding authors.

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

## Acknowledgements

The authors acknowledge the medical writing assistance of Anne-Marie Eades-Perner (Charles River Discovery Research Services Germany GmbH), for the revisions and formatting of the manuscript. The authors would also like to thank Stefanie Klingel and Sandra Ribeiro (Charles River Discovery Research Services Germany GmbH) for the maintenance of the sample biobank, the preparation of the DNA and RNA samples and the qPCR experiments done, and Serenella Eppenberger-Castori (Uninversity of Basel) for the MLH1 immunohistochemistry of the PDX.

## Author contributions

Conception and design were performed by C.R., A.A., V.V., H.-K.Y., and H.-H.F. The development of methodology was done by A.-L.P., S.-S.W., I.W., S.B., J.T., and W.H.K. The data (provided animals, acquired and managed patients, provided facilities, etc.) was acquired by A.-L.P., V.V., K.K., P.B., W.H.K., H.-J.L., S.-H.K., and H.-K.Y. The analysis and interpretation of data (e.g., statistical analysis, biostatistics, computational analysis) was conducted by A.-L.P., S.-S.W., M.L., B.Z., A.A., J.T., and V.V. Writing, review, and/or revision of the manuscript was done by A.-L.P., S.-S.W., A.A., and V.V. N.Z., G.D., C.R., W.H.K., H.-J.L., and S.-H.K. provided administrative, technical, or material support (i.e., reporting or organizing data, constructing databases). B.Z., M.L., S.-S.W., J.T., I.W., S.B., A.A., and V.V. analyzed all the molecular data used in this paper. A.A., V.V., C.R., and H.-H.F. supervised the study.

## Competing interests

N.Z. and K.K. are employed by Charles River Discovery Research Services Germany GmbH (formerly Oncotest). H.-H.F., V.V., and A.-L.P. were employed by Charles River Discovery Research Services Germany GmbH (formerly Oncotest) and are now employed by 4HF Biotec GmbH. S.-S.W. and S.B. were employees and stockholders of Eli Lilly and Company and are now employees of LifeOmic. I.W., J.T., A.A., G.D., and C.R. are stockholders and employees of Eli Lilly and Company. B.Z. and M.L. were employed by Charles River Discovery Research Services Germany GmbH (formerly Oncotest) and are now employed by Sogeti and Evotec International GmbH, respectively. The other authors have no competing interests with the present study.
