## [Peer Review File · Communications Biology]

Reviewers' comments:

Reviewer #1 (Remarks to the Author):

This is an interesting report on evaluating factors that affect establishment of GC-PDX model from primary cancer patients. In consistence with previous studies that PDX well preserved histology, genomic and transcriptome subtypes of parental tumors. Furthermore, this study revealed that genetic changes are selected at early passages and rare subclones can emerge in PDX. However, the biggest problem is that author didn't clearly explain the reasons why the engraftment percentage of intestinal, MSI, MSS/TP53-, MSS/TP53+, diffuse, EBV+ and EMT tumors subtypes are different.

Major Essential Revision Points 1) The author claims that "For using of GC-PDX, histology is not sufficient, it requires detailed knowledge of their molecular characteristics". The author should tell researchers which molecular characteristics of GC cancer (RNA-seq/WGS et al.) must be done for screening compounds or other tests?
2) The author did not confirm biologic functions of the bioinformatics results of GC and GC-PDX cells.
3) The results didn't support title that Molecular subtypes and clonal selection influence the establishment of GC-PDX".

Reviewer #2 (Remarks to the Author):

Peille et al. established 27 gastric cancer PDX models from 100 tumors resected for Asian cancer patients. They performed genomic characterization of the new models, and compared their genomic profiles to those from patients' data, finding that some molecular subtypes are better represented than others. Next, they followed the clonal evolution of PDXs by comparing primary tumor samples to consecutive PDX passages, and found evidence for early selection that leads to the expansion of rare tumor subclones.

The conceptual novelty of this study is limited, in light of recent literature, and especially two high-profile papers that made similar observations and reached similar conclusions (both are cited in the manuscript: Eirew et al. Nature 2015; Ben-David et al. Nature 2017). However, the discussion surrounding the stability and clonal selection of PDX models remains very lively, making this manuscript both timely and relevant. Additionally, the establishment and profiling of 27 gastric cancer PDXs (out of 100 attempts) is no small feat, and the use of ddPCR to demonstrate the selection of pre-existing genetic alterations is very nice.

Below are a few suggestions for modifications and additional analyses that could strengthen the conclusions of the study, as well as its potential impact:

1) A key point of this study is that the GC PDX cohort is skewed toward specific GC subtypes. However, not all of the claims are backed up by appropriate statistical tests:
Lines 78-80: "PDX models were mainly obtained from MSI tumors (n=15/27), MSS/TP53- (n=9/32) and MSS/TP53+ (3/21) but not EMT and EBV+ positive tumors." Is this statistically significant?
Lines 96-98: "We observed that the collection had very similar distribution of these alterations than those observed across TCGA tumors." What does 'similar' mean? A statistical comparison of the two cohorts should be performed.

2) The finding that in all MSI PDX models the copy number of chromosome 7 increased is quite intriguing. Is the number of samples sufficient for calculating the statistical significance of this

observation? Can the authors speculate what might underlie the recurrence of this particular chromosome gain?

3) The authors' data convincingly show copy number instability (in MSS models) and mutational instability (in MSI models) throughout the engraftment and early propagation of GC PDXs. It makes a lot of sense that clonal evolution throughout PDX establishment and propagation would mostly occur through the cellular mechanism that is already 'defective' to begin with. In line with this, is chromosomal instability also higher in MSS/TP53-mutant PDXs compared to MSS/TP53-WT PDXs?

4) At least some of the extensive genetic evolution could be attributed to the expansion of rare subclones that already existed in the patient's tumor, as demonstrated by the ddPCR analysis. These conclusions align well with those obtained by Ben-David et al., who used LOH analysis of several PDX models to reach an identical conclusion, and with those obtained by Eirew et. al., who used single cell sequencing of breast cancer PDX models. The agreement between the studies should be explicitly mentioned (it is currently stated in a rather vague manner in lines 166 and 179-181). For example, the authors state that "our study is largely based on one tumor type, and findings from here may not extrapolate to other tumor and host types" (lines 185-186), but the main observations have actually been previously made in other tumor types and this should be directly acknowledged.

5) For the mutations that present an increase in their allelic fraction at P1, the authors propose that this "is probably due to a clonal election in the passages and/or removal of human stromal cells" (lines 134-135). Is it possible to distinguish between these possibilities? If the observed increase in AF is due to the removal of human stromal cells, then all mutations with similar AF should similarly 'expand'; if it's clonal selection, however, then some mutations would expand whereas others would retain their AF throughout passaging. Can this prediction be useful in resolving the proposed ambiguity?

6) The manuscript would benefit from language editing.

Reviewer #3 (Remarks to the Author):

The authors compiled a collection of 100 gastric adenocarcinoma patient samples. Serial passages in mice revealed 27 samples that were subsequently sequenced and could be used for future therapeutic testing. This work adds to previous community effort to assemble a collection of PDX samples that represent disease landscape.

It would be useful if the authors summarized in a table how many more were transplanted with each patient sample at each passage

To better understand the data I would suggest the authors to work on better presentation of data in tables (including table 1) possibly make supplemental tables sortable (spreadsheets).

Please indicate on which conditions described PDX models are available to outside researchers.

Minor comments:

Figure S1A is too complex not very informative or needed for the conclusion.

Figure S5A-D please make plots square

Reviewers' comments:

Reviewer #1 (Remarks to the Author):

This is an interesting report on evaluating factors that affect establishment of GC-PDX model from primary cancer patients. In consistence with previous studies that PDX well preserved histology, genomic and transcriptome subtypes of parental tumors. Furthermore, this study revealed that genetic changes are selected at early passages and rare subclones can emerge in PDX. However, the biggest problem is that author didn't clearly explain the reasons why the engraftment percentage of intestinal, MSI, MSS/TP53-, MSS/TP53+, diffuse, EBV+ and EMT tumors subtypes are different.

Major Essential Revision Points:

1) The author claims that "For using of GC-PDX, histology is not sufficient, it requires detailed knowledge of their molecular characteristics". The author should tell researchers which molecular characteristics of GC cancer (RNA-seq/WGS et al.) must be done for screening compounds or other tests?

As pointed out by the reviewer, this statement in discussion section doesn't build on results pertaining to screening of compounds. We merely wanted to state that heterogeneity of parental tumor.

Original text from discussion: "This study has important implications. Firstly, the consideration of GC PDX models on the simple basis of their type or histology is not sufficient, it requires detailed knowledge of their molecular characteristics for a proper use. The commonly held belief that PDX reflect parental tumors, needs to be adjusted in the context of this data and other recently emerging data suggestive of inadequacies in PDX models. Both, establishment bias and clonal selection during GC PDX establishment happen making these models differ both at GC population level as well as at the level of parental tumors"

Modified text: "This study has important implications. Firstly, commonly held belief that PDX reflect parental tumors, needs to be adjusted in the context of this data and other recently emerging data suggestive of inadequacies in PDX models. Both, establishment bias and clonal selection during GC PDX establishment can happen making these models differ both at GC population level as well as at the level of parental tumors. This study shows that the consideration of GC PDX models on the simple basis of their type or histology is not sufficient, it requires detailed knowledge of their molecular characteristics for a proper use. An extensive molecular characterization, in terms of gene mutations, gene expression and gene copy number, is suitable for an appropriate use of these PDX for drug testing experiments. A careful selection of PDX models based on their molecular pattern may increase the success of drug testing experiments and/or may allow to identify molecular determinate of the sensitivity response."

2) The author did not confirm biologic functions of the bioinformatics results of GC and GC-PDX cells.

Apologies, this comment is not clear. Could you kindly provide us with more information about this comment?

3) The results didn't support title that Molecular subtypes and clonal selection influence the establishment of GC-PDX".

As per the suggestion, we have changed the title to “Evaluation of molecular subtypes and clonal selection during establishment of patient-derived tumor xenografts from Asian gastric adenocarcinoma”

4) the biggest problem is that author didn't clearly explain the reasons why the engraftment percentage of intestinal, MSI, MSS/TP53-, MSS/TP53+, diffuse, EBV+ and EMT tumors subtypes are different.

We modified the first part of the discussion to include our hypotheses regarding the preferential engraftment of the MSI and CIN tumors (particularly ERBB2 amplified) over the genomic stable and EBV+ tumors. The MSI tumors accumulate a high number of mutations. This characteristic may confer a certain adaptability to the tumor cells and thus a facility to grow in a new microenvironment (the immunocompromise mice). PDX were also frequently established from ERBB2+ tumors, probably because of the capacity of these cells to proliferate without the expression of the corresponding ligand (ligand independent growth). Other subtypes, such as the genomic stable, may require additional growth factors to proliferate which might not be available in the immunocompromise mice.

Reviewer #2 (Remarks to the Author):

Peille et al. established 27 gastric cancer PDX models from 100 tumors resected for Asian cancer patients. They performed genomic characterization of the new models, and compared their genomic profiles to those from patients' data, finding that some molecular subtypes are better represented than others. Next, they followed the clonal evolution of PDXs by comparing primary tumor samples to consecutive PDX passages, and found evidence for early selection that leads to the expansion of rare tumor subclones.

The conceptual novelty of this study is limited, in light of recent literature, and especially two high-profile papers that made similar observations and reached similar conclusions (both are cited in the manuscript: Eirew et al. Nature 2015; Ben-David et al. Nature 2017). However, the discussion surrounding the stability and clonal selection of PDX models remains very lively, making this manuscript both timely and relevant. Additionally, the establishment and profiling of 27 gastric cancer PDXs (out of 100 attempts) is no small feat, and the use of ddPCR to demonstrate the selection of pre-existing genetic alterations is very nice.

Below are a few suggestions for modifications and additional analyses that could strengthen the conclusions of the study, as well as its potential impact:

1) A key point of this study is that the GC PDX cohort is skewed toward specific GC subtypes. However, not all of the claims are backed up by appropriate statistical tests:

- Lines 78-80: “PDX models were mainly obtained from MSI tumors (n=15/27), MSS/TP53- (n=9/32) and MSS/TP53+ (3/21) but not EMT and EBV+ positive tumors.” Is this statistically significant?

We have established GC PDX more frequently from MSI GC patient tumors than MSS GC patient tumors ($p < 0.0001$, table 1). None of the EBV GC patient tumors were established as PDX ($p = 0.10$, see table 1). Only one MSS/EMT GC patient tumor was successfully established as PDX but was classified as MSS/TP53-.

We have modified the text to: "PDX models were predominantly obtained from MSI tumors (n=15/27, p<0.0001, see table 1), followed by MSS/TP53- (n=9/32) and MSS/TP53+ (3/21) but not EMT and EBV+ positive tumors." And included the results of the statistical test to make this statement stronger.

- Lines 96-98: "We observed that the collection had very similar distribution of these alterations than those observed across TCGA tumors." What does 'similar' mean? A statistical comparison of the two cohorts should be performed.

By 'similar' we meant that the distributions of the alterations in the PDX and in the patient tumors are correlated. We have compared the PDX to the patient tumors (TCGA cohort) in terms of alteration frequency in the 48 targetable genes, for the MSI and the CIN subtypes. For the MSI PDX and patient tumors, we calculated a Spearman correlation of 0.67 and p<0.0001. For the CIN PDX and patient tumors, the Spearman correlation was at 0.55 with p<0.0001. The GS and EBV subtypes were not included in this comparison due to the low number or absence of PDX models established for these molecular subtypes. We have included the results of the statistical tests in the text to make our statement clearer.

2) The finding that in all MSI PDX models the copy number of chromosome 7 increased is quite intriguing. Is the number of samples sufficient for calculating the statistical significance of this observation? Can the authors speculate what might underlie the recurrence of this particular chromosome gain?

It is a little complicated to identify a common driver of this genomic gain due to the low number of MSI PDX studied. A larger cohort is necessary to investigate this change in more detail. However, we can provide the following additional information. A small increase of the allelic fraction (AF) is observed in all 4 MSI PDX model. It increases from 0.0046 (range 0.0034 to 0.0053) in the normal tissue to 0.096 (range 0.07 to 0.11) in the patient tumors, to 0.24 (range 0.22 to 0.28) at P1 and remains stable at P2, P3 and in the established PDX. Nevertheless, a certain heterogeneity can be observed with genes showing a stable AF, genes with a moderate increase (approx. up to 0.3) and few genes with a high increase (between 0.6 and 1). Among the genes located on the chromosome 7 and presenting a genomic gain, we can mention genes related to the mismatch repair, SHH or WNT pathways for example (e.g; PMS2, POLM, EGFR, MET, SMO, SHH).

3) The authors' data convincingly show copy number instability (in MSS models) and mutational instability (in MSI models) throughout the engraftment and early propagation of GC PDXs. It makes a lot of sense that clonal evolution throughout PDX establishment and propagation would mostly occur through the cellular mechanism that is already 'defective' to begin with. In line with this, is chromosomal instability also higher in MSS/TP53-mutant PDXs compared to MSS/TP53-WT PDXs?

Among the MSS/TP53- (GXA_3012, GXA_3013, GXA_3029, GXA_3038, GXA_3039, GXA_3040, GXA_3063, GXA_3067 and GXA_3084), only one PDX (GXA_3012) is TP53 wild-type.

*GXA_3013_Cys238Phe,
GXA_3029_Arg273Cys,
GXA_3038_Gln104*,*

GXA_3039_Val272Met,
GXA_3040_Lys132Asn & X124X,
GXA_3063_Trp146*,
GXA_3067_X200X,
GXA_3084_Ala189Val & Trp146*

This GC PDX has 155 SCNA (52 gene amplifications and 103 gene deletions). GXA_3012 has the highest number of gene deletions among the MSS/TP53- GC PDX (range: 5 to 103, and 5 to 79 when the GXA_3012 is excluded). The number of gene amplifications is low in this PDX model (52, range of the subtype: 0 to 250), but is not different from the other MSS/TP53- GC PDX.

4) At least some of the extensive genetic evolution could be attributed to the expansion of rare subclones that already existed in the patient's tumor, as demonstrated by the ddPCR analysis. These conclusions align well with those obtained by Ben-David et al., who used LOH analysis of several PDX models to reach an identical conclusion, and with those obtained by Eirew et. al., who used single cell sequencing of breast cancer PDX models. The agreement between the studies should be explicitly mentioned (it is currently stated in a rather vague manner in lines 166 and 179-181). For example, the authors state that "our study is largely based on one tumor type, and findings from here may not extrapolate to other tumor and host types" (lines 185-186), but the main observations have actually been previously made in other tumor types and this should be directly acknowledged.

The reason why we used this as a potential shortcoming is that Eirew et al used breast cancer models whereas Ben-David et al only showed that in context of cell lines. We have modified the wordings as per the suggestion of the reviewer. Discussion: "Fourthly, our study is largely based on one tumor type and findings from here may not extrapolate to other tumor and host types" has now been modified to: "Fourthly, our study is focused on one tumor type. Despite similar findings observed in cell lines and breast cancer models (Eirew & Ben-David), the extensive characterization of PDX established from other tumor entities is needed to determine if these findings can be extrapolated to all tumor types."

5) For the mutations that present an increase in their allelic fraction at P1, the authors propose that this "is probably due to a clonal selection in the passages and/or removal of human stromal cells" (lines 134-135). Is it possible to distinguish between these possibilities? If the observed increase in AF is due to the removal of human stromal cells, then all mutations with similar AF should similarly 'expand'; if it's clonal selection, however, then some mutations would expand whereas others would retain their AF throughout passaging. Can this prediction be useful in resolving the proposed ambiguity?

We thank the reviewer for their astute observation. Depending on the PDX model any of the mechanism could be at play. For e.g. Figure 2, GXA_3037 shows a case of clonal selection at passage 1 (note the change in AF in lower panel, where some mutations "expanded" as seen by AF increase). In the same Figure 2, GXA3040 where an increase is noted the AF suggestive of stroma removal. The statement above represents what we observed across PDX models and not specific to any one model. We have changed the wording to give examples of both in the text to make this clearer for the reviewer and the reader.

6) The manuscript would benefit from language editing.

As per the suggestion we have undergone an additional review and made changes.

Reviewer #3 (Remarks to the Author):

The authors compiled a collection of 100 gastric adenocarcinoma patient samples. Serial passages in mice revealed 27 samples that were subsequently sequenced and could be used for future therapeutic testing. This work adds to previous community effort to assemble a collection of PDX samples that represent disease landscape.

1) It would be useful if the authors summarized in a table how many more were transplanted with each patient sample at each passage.

A total of 100 GC patient tumors were implanted and 73 did not go beyond passage 1.

2) To better understand the data I would suggest the authors to work on better presentation of data in tables (including table 1) possibly make supplemental tables sortable (spreadsheets).

The online submission likely converted the tables to a pdf file. As per the suggestion, we have made the supplemental tables available in Microsoft xls.

3) Please indicate on which conditions described PDX models are available to outside researchers.

PDX models are available to researchers for drug testing from Charles River Laboratories as a fee for service.

Minor comments:

4) Figure S1A is too complex not very informative or needed for the conclusion.

We agree with the comment of the reviewer. This figure is not needed for the conclusion, thus we included it as a supplementary figure. Our goal with this figure and the attached supplementary information was to provide an exhaustive characterization of the PDX in terms of genomic instability. We can of course simplify it (by removing the parts C and D for example) if the reviewer prefers.

5) Figure S5A-D please make plots square

We have generated a new figure S5 with square plots.

REVIEWERS' COMMENTS:

Reviewer #1 (Remarks to the Author):

The authors collected 100 GC tumor tissues resected from Asian patients. After series passage in mice, only 27 PDX with molecular subtype bias (MSI, CIN and MSS/TP53-) were stably established. In addition, they identified that genetic changes are selected at early passages, and subsequently, mutational signature and subclones are different between parental tumors and PDX. They claimed that GC PDX collection cannot fully cover the diversity of gastric cancers so molecular subtypes and possible clonal evolution should be an important consideration for various translational studies. This manuscript tells us that we need more considerations to ensure that the chosen PDX model for drug testing is accurate and appropriate. However, some points are worth being discussed or improved. First of all, in this manuscript, it's unclear what's the tumor volume when tumor tissues were transplanted into mice. How many mice were prepared for each sample and how many tumor tissues were transplanted in one mouse?

When the authors evaluated molecular subtypes and subclones, did they evaluate multiple Pn+1 samples collected from different mice but derived from the same Pn sample? As known, tumor tissues are heterogeneous. When cutting and transplanting, the tissue pieces separated from the same sample might be already different in molecular subtypes and subclones.

Similarly, is there a possibility that there are already some differences between the primary tumors being tested and the primary tumors used for transplantation? Repeated experiments may avoid these errors and reduce this concern. So please tell us the details of these experiments and take these concerns into account.

In addition, this manuscript needs further editing. For example, it would be better to show its full spelling after the abbreviations for the first time. Finally, I believe that " Firstly, we observed that tumors of the Lauren's intestinal subtype were established more commonly than diffuse or mixed tumors, confirming previously published data 7, 10, 9, 12, 11. " (line 179) could be a typo.

Reviewer #2 (Remarks to the Author):

The authors have adequately addressed most of my concerns. Please see a couple of minor clarifications that are still required:

"We have modified the text to: "PDX models were predominantly obtained from MSI tumors (n=15/27, p<0.0001, see table 1), followed by MSS/TP53- (n=9/32) and MSS/TP53+ (3/21) but not EMT and EBV+ positive tumors." And included the results of the statistical test to make this statement stronger."

Is this the p-value for MSI tumors specifically, or is this the result of a Fisher's Exact test of the full (2x5) contingency table? It's not clear exactly which statistical test was performed here.

"The reason why we used this as a potential shortcoming is that Eirew et al used breast cancer models whereas Ben-David et al only showed that in context of cell lines. We have modified the wordings as per the suggestion of the reviewer. Discussion: "Fourthly, our study is largely based on one tumor type and findings from here may not extrapolate to other tumor and host types" has now been modified to:

"Fourthly, our study is focused on one tumor type. Despite similar findings observed in cell lines and

breast cancer models (Eirew & Ben-David), the extensive characterization of PDX established from other tumor entities is needed to determine if these findings can be extrapolated to all tumor types.”

This statement is inaccurate. The paper by Ben-David et al. (28991255) showed this phenomenon in multiple types of PDXs (rather than in cell lines). Additional papers also reported similar results in other tumor types (e.g., in pancreatic cancer; PMID 30629588)

Reviewer #3 (Remarks to the Author):

I am satisfied with Authors responses, the work was greatly improved in resubmission. I do not have further questions

REVIEWERS' COMMENTS:

Reviewer #1 (Remarks to the Author):

The authors collected 100 GC tumor tissues resected from Asian patients. After series passage in mice, only 27 PDX with molecular subtype bias (MSI, CIN and MSS/TP53-) were stably established. In addition, they identified that genetic changes are selected at early passages, and subsequently, mutational signature and subclones are different between parental tumors and PDX. They claimed that GC PDX collection cannot fully cover the diversity of gastric cancers so molecular subtypes and possible clonal evolution should be an important consideration for various translational studies. This manuscript tells us that we need more considerations to ensure that the chosen PDX model for drug testing is accurate and appropriate. However, some points are worth being discussed or improved. First of all, in this manuscript, it's unclear what's the tumor volume when tumor tissues were transplanted into mice.

Tumor pieces of 1-2 mm³ were transplanted into mice. We have included this information in the manuscript.

How many mice were prepared for each sample and how many tumor tissues were transplanted in one mouse?

Two tumor pieces were implanted per mouse, bilaterally into the flank of immunodeficient mice. For the genomic analyses of PDX, one piece taken from one PDX was used to generate a DNA sample that was further used to perform gene copy number analysis and whole exome sequencing. For the transcriptomic analyses of PDX, 4 PDX pieces of the same model (one piece from four PDX tumors) were pooled before preparing the RNA samples. This pooling was done to limit the effect of the inter/intra-tumor variability. This information was included in the methods section.

When the authors evaluated molecular subtypes and subclones, did they evaluate multiple Pn+1 samples collected from different mice but derived from the same Pn sample? As known, tumor tissues are heterogeneous. When cutting and transplanting, the tissue pieces separated from the same sample might be already different in molecular subtypes and subclones. Similarly, is there a possibility that there are already some differences between the primary tumors being tested and the primary tumors used for transplantation? Repeated experiments may avoid these errors and reduce this concern. So please tell us the details of these experiments and take these concerns into account.

Thanks for raising an important concern. We did not plan for evaluation of multiple Pn+1 samples for genomics data generation for a given PDX model as we only had limited budget for this study. We have only evaluated one sample per condition.

We totally agree with the reviewer comment, tumors are heterogeneous and the cutting of tumor pieces before implantation into the mice or before DNA/RNA preparation may introduce a bias. As mentioned briefly in the first limitation in discussion, "it is possible that the selection of tumor fragment for establishment and characterization, degree of immuno-deficiency in the mouse strain (for e.g. SCID vs NSG) and the environmental context (lack of certain growth factors) may influence the establishment rate as well as clonal selection".

It is possible that based on the tumor fragment utilized a different subclones may arise. As shown through ddPCR experiments, these clones likely were very rare ones that emerged during passaging. Emergence of rarer subclones during early passaging may happen but the end result and for e.g. presence of a particular mutation would be susceptible to the choice of fragment used. Furthermore, it is also possible that using multiple fragments from parental tumors could help establish more PDXs for e.g. for GS or EMT subtype, which tends to have higher stromal content. It is also possible that use of a different mouse strain (for e.g. SCID vs NSG) may have helped establish EBV positive gastric cancers. There could however be environmental factors such as lack of certain

growth factors, which could have contributed to the results observed and have not been systematically studied. We cannot exclude that the difference observed between the samples are due to the pieces of tumors analyzed. This experiment was conceived as a screen. We agree that a deeper analysis with multiple samples per condition/passage is needed to reinforce the statement about the clonal selection and tumor evolution.

However, the molecular subtypes are defined by the presence of key features such as the EBV positivity, MSI status, large chromosomal rearrangements. The persistence of these characteristics in the individual PDX shows that the piece used for the analyses well represents the original patient tumor in terms of key molecular features. Also, when we compared the genomic profiles of PDX across passages (Fig. 6c) we analyzed large changes. In the Figure 6d, we only considered mutations with 10 read coverage in all samples of the same model to avoid bias linked to the analysis of single tumor pieces.

In addition, this manuscript needs further editing. For example, it would be better to show its full spelling after the abbreviations for the first time. Finally, I believe that " Firstly, we observed that tumors of the Lauren's intestinal subtype were established more commonly than diffuse or mixed tumors, confirming previously published data 7, 10, 9, 12, 11. " (line 179) could be a typo.

We have reduced the number of abbreviations and included the full spelling after the abbreviations for the first time. We have corrected the typo on line 179.

We sincerely thank the reviewer #1 for these questions that helped us to improve the quality of our paper.

Reviewer #2 (Remarks to the Author):

The authors have adequately addressed most of my concerns. Please see a couple of minor clarifications that are still required:

"We have modified the text to: "PDX models were predominantly obtained from MSI tumors (n=15/27, p<0.0001, see table 1), followed by MSS/TP53- (n=9/32) and MSS/TP53+ (3/21) but not EMT and EBV+ positive tumors." And included the results of the statistical test to make this statement stronger."

Is this the p-value for MSI tumors specifically, or is this the result of a Fisher's Exact test of the full (2x5) contingency table? It's not clear exactly which statistical test was performed here.

We did a Fisher's exact test comparing the PDX success rate in the MSI vs the MSS categories and obtained a p-value of 0.0002. Sorry there was a typo in the p-value indicated and we have corrected it.

Data indicated in the table 1:

MSI status	Number of patient tumors	Number of PDX established	Success rate (%)
MSI	27 (27%)	15 (56%)	56
MSS	73 (73%)	12 (44%)	16
Unknown	0 (0%)	0 (0%)	0

The contingency table used for the Fisher's exact test was:

	Number of patient tumors established as PDX	Number of patient tumors not established as PDX
MSI	15	12 (=27-12)
MSS	12	61 (=73-12)

We have changed the text as follow: "In line with previous studies^{17,18}, the PDX collection was enriched in models established from MSI tumors (n=15/27, Fisher's exact test, p<0.0001, Table 1). A

lower number of PDX were developed from MSS/TP53- (n=9/32) and MSS/TP53+ (3/21), but none from EMT and EBV positive tumors.”

“The reason why we used this as a potential shortcoming is that Eirew et al used breast cancer models whereas Ben-David et al only showed that in context of cell lines. We have modified the wordings as per the suggestion of the reviewer. Discussion: “Fourthly, our study is largely based on one tumor type and findings from here may not extrapolate to other tumor and host types” has now been modified to:

“Fourthly, our study is focused on one tumor type. Despite similar findings observed in cell lines and breast cancer models (Eirew & Ben-David), the extensive characterization of PDX established from other tumor entities is needed to determine if these findings can be extrapolated to all tumor types.” This statement is inaccurate. The paper by Ben-David et al. (28991255) showed this phenomenon in multiple types of PDXs (rather than in cell lines). Additional papers also reported similar results in other tumor types (e.g., in pancreatic cancer; PMID 30629588)

Apologies for this error. There are two Ben-David publications.

A) <https://www.ncbi.nlm.nih.gov/pubmed/28991255> (PDX)

B) <https://www.ncbi.nlm.nih.gov/pubmed/30089904> (Cell lines).

Publication A had used copy number data along with inferred copy number data without any mutational analysis across passages whereas B) in cell lines was more comprehensive and that’s what we were referring to. The papers published by Gendoo et al, and by Eirew et al, focus on the genomic profile of patient tumors, PDX (either established or at early passages) and PDO. The paper published by Corso et al, demonstrates the preservation of key characteristics of gastric PDX at the “collection level”.

This was a comment on potential shortcomings, which we were perhaps being overly critical. The changes read as follows. *“Fourthly, our study is focused on one tumor type; However, similar findings have been observed in cell lines and PDX (Eirew, Ben-David, Ben-David, Gendoo, Corso), likely suggesting the existence and an important consideration in other tumor types and associated tumor model system.”*

We thank the reviewer #2 for bringing this interesting paper about pancreatic PDX (Gendoo) to our attention.

We hope the rephased sentence suits the reviewer #2.

Reviewer #3 (Remarks to the Author):

I am satisfied with Authors responses, the work was greatly improved in resubmission. I do not have further questions

We thank the reviewer #3 for his feedback and for time taken to review the article.